# HYDEN: A HYBRID DUAL-PATH ENCODER FOR MONOCULAR GEOMETRY OF HIGH-RESOLUTION IMAGES

**Zaiwei Zhang & Marc Mapeke & Wei Ye & Rakesh Ranjan & JQ Huang**
Meta Reality Labs
322 Airport Blvd.
Burlingame, CA 94010, USA
`{zaiweizhang,mmapeke,weiye,rakeshr,jqhuang}@meta.com`

## ABSTRACT

We present a **hy**brid **d**ual-path vision **en**coder (Hyden) for high-resolution monocular depth, point map and surface normal estimation, surpassing state-of-the-art accuracy with a fraction of the inference cost. The architecture pairs a low-resolution Vision Transformer branch for global context with a full-resolution CNN branch for fine details, fusing features via a lightweight MLP before decoding. By exploiting the linear scaling of CNNs and constraining transformer computation to a fixed resolution, the model delivers fast inference even on multi-megapixel inputs. To overcome the scarcity of high-quality high-resolution supervision, we introduce a self-distillation framework that generates pseudo-labels from existing models at both lower resolution full images and high-resolution crops—global labels preserve geometric accuracy, while local labels capture sharper details. To demonstrate the flexibility of our approach, we integrate Hyden and our self-distillation method into DepthAnything-v2 for depth estimation and MoGe2 for surface normal and metric point map prediction, achieving state-of-the-art results on high-resolution benchmarks with the lowest inference latency among competing methods.

## 1 INTRODUCTION

Monocular depth, pointmap, and surface normal estimation are core to 3D perception in driving, robotics, and mixed reality. Models like MiDaS Ranftl et al. (2020) and DepthAnything Yang et al. (2024) show strong results from a single RGB image, but most are trained at low resolution, causing degraded predictions on megapixel inputs Wang et al. (2025).

To close this gap, recent work partitions images into tiles and blends ViT features (Depth-Pro Bochkovskii et al. (2024), PatchFusion Li et al. (2024b)) or designs multi-branch ViTs (FlashDepth Chou et al. (2025)). Yet, ViT inference scales quadratically with resolution. Supervision is also problematic: real high-res supervisions are often noisy or sparse, while synthetic labels are perfect but introduce domain gaps, making model generalization difficult.

To address these issues we present Hyden—a Hybrid Dual-path Encoder coupling a full-resolution CNN with a low-resolution ViT. CNN features preserve local detail, while upscaled ViT tokens provide global context, fused through lightweight layers before task-specific decoding. This design substantially lowers inference latency while maintaining sharp predictions.

For supervision, Hyden uses self-distillation: unlabeled high-resolution images are pseudo-labeled by a frozen teacher at $518 \times 518$ for both full images and high-res crops. The original ViT branch is kept frozen, and only the CNN branch, fusion layer, and decoder are optimized using both a global loss (on the downsampled full image) and a local crop loss (on masked crop regions).

Our contributions is summarized as follows:

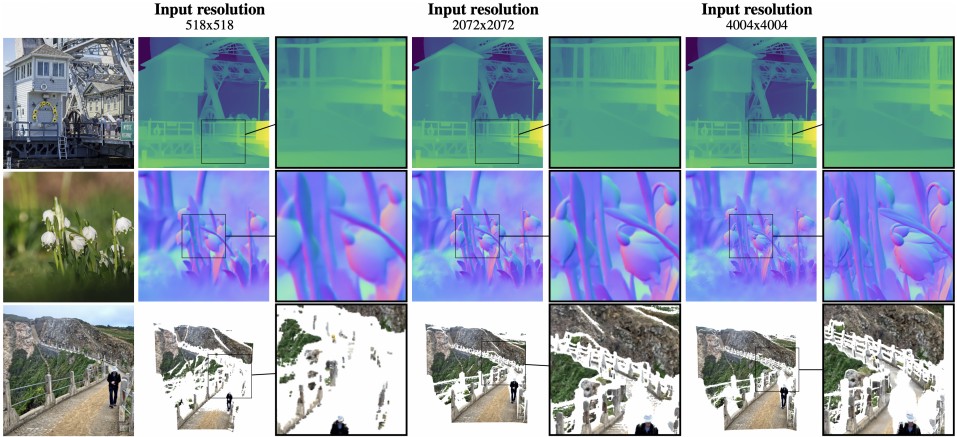

(a) Example inferences with Hyden-DA2 (row 1), Hyden-MoGe2-Normal (row 2), Hyden-MoGe2-Pointmap (row 3) across inputs with different resolutions

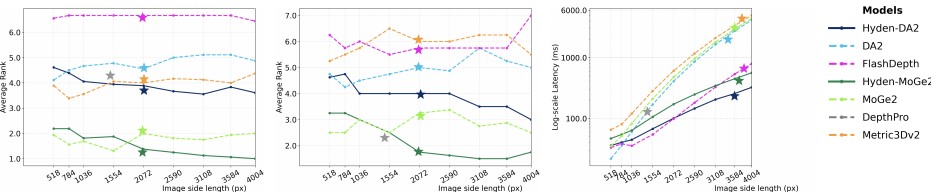

(b) Geometry Accuracy (↓)   (c) Geometry Sharpness (↓)   (d) Latency (log-scale) (↓)

Figure 1: **Performance comparison across inference resolutions for geometric foundation models:** (a) Example inferences with relative depth, surface normal and point map prediction models illustrate the tradeoff between latency and sharpness across resolutions. (b) shows average ranking across datasets for relative depth accuracy, (c) reports average ranking for geometry sharpness across depth models, and (d) plots inference latency (**log-scale**) measured on an NVIDIA A100 GPU with FP16 precision. **Lower is better for all plots.** Compared to base models, both Hyden-DA2 and Hyden-MoGe2 achieve improved accuracy at high-resolution inference and deliver significant inference speedups. Hyden-MoGe2 achieves the best geometry accuracy and sharpness compared to other state-of-the-art models and consumes significantly lower inference latency. (*DepthPro is evaluated at a fixed resolution due to model constraint.)

- We introduce Hyden, the first encoder that combines a fixed-resolution ViT for global context and a full-resolution CNN for fine detail, significantly reducing inference cost while preserving high-resolution accuracy.

- We propose a self-distillation framework that uses global pseudo-labels to preserve accurate geometry across resolutions and local pseudo-labels to capture sharper fine details in high-resolution predictions.

- By integrating the Hyden encoder into two leading models: DepthAnything-v2 Yang et al. (2024) and MoGe2 Wang et al. (2025)—our approach establishes new state-of-the-art performance for high-resolution depth, point map, and surface-normal prediction, while achieving average 3x lower inference latency at 2K and nearly 10× lower at 4K resolution compared to the original models (see Figure 1).

## 2 RELATED WORK

### 2.1 ZERO-SHOT MONOCULAR GEOMETRY ESTIMATION

Traditional monocular models Bhat et al. (2021); Eigen et al. (2014); Li et al. (2022); Eigen & Fergus (2015); Saxena et al. (2008) were trained on single datasets for specific domains (e.g., indoor or street-view) and generalized poorly due to limited diversity and fixed camera setups.

**Relative depth** To improve generalization, MegaDepth Li & Snavely (2018) and DiverseDepth Yin et al. (2020) scaled supervision with Internet-scale data. MiDaS Ranftl et al. (2020) introduced scale- and shift-invariant losses, later extended with transformers Ranftl et al. (2021); Birkl et al. (2023). DepthAnything Yang et al. (2024) distilled pseudo labels for 62M images, while generative priors adapted diffusion models Ke et al. (2024); Rombach et al. (2022) or joint attention Fu et al. (2024). These methods generalize broadly but remain limited by scale/shift ambiguity.

**Metric depth** Scarce metric annotations hinder absolute scale. ZoeDepth Bhat et al. (2023) fine-tunes metric heads on relative models. Metric3D Hu et al. (2024) and DepthPro Bochkovskii et al. (2024) resolve cross-camera ambiguity via canonical transformations. UniDepth Piccinelli et al. (2025) learns implicit camera models, while MoGe2 Wang et al. (2025) predicts scale-invariant pointmaps with scale recovery.

**Metric point map** Another direction is predicting 3D pointmaps. Many works Yin et al. (2021); Piccinelli et al. (2025); Hu et al. (2024); Bochkovskii et al. (2024) decouple depth and camera recovery, e.g., LeRes Yin et al. (2021) regresses depth and intrinsics, UniDepth Piccinelli et al. (2024) uses camera embeddings. DUSt3R Wang et al. (2024) predicts stereo pointmaps end-to-end, and MoGe2 Wang et al. (2025) combines scale-invariant maps with scale factors.

Despite progress, most models are trained at low resolution ($\leq 518 \times 518$), losing detail when downsampled and incurring high cost at full scale.

## 2.2 Zero-shot Surface Normal Estimation

Normals avoid metric ambiguity and capture local shape for localization Behley & Stachniss (2018), mapping Wang et al. (2019), and reconstruction Yu et al. (2022); Wang et al. (2022). Early work derived them from RGB-D scans Silberman et al. (2012); Eigen & Fergus (2015); Qi et al. (2020) and denoised via consistency Qi et al. (2018), adaptive constraints Long et al. (2024; 2021), or uncertainty Bae et al. (2021). OmniData Eftekhar et al. (2021) scaled to 1.3B frames, while Normal-in-the-Wild Chen et al. (2017) expanded to outdoor scenes. DSINE Bae & Davison (2024) introduced a normals-specific architecture, and recent transformer-based approaches Hu et al. (2024); Wang et al. (2025) unify depth, normals, and pointmaps. Yet most remain constrained to low-resolution training, reducing sharpness.

## 2.3 High-Resolution Depth and Surface Normal Estimation

To recover fine details, SMD-Net Tosi et al. (2021) and Poisson-fusion Dai et al. (2023); Li et al. (2024a) sharpen boundaries, while patch pipelines—BoostingDepth Miangoleh et al. (2021), PatchFusion Li et al. (2024b), PatchRefiner Li et al. (2024c)—boost local detail but introduce artifacts and latency. PRO Kwon & Kim (2025) cuts computation but lags end-to-end models Bochkovskii et al. (2024); Chou et al. (2025). DepthPro Bochkovskii et al. (2024) improves patch efficiency, and FlashDepth Chou et al. (2025) uses dual-branch ViTs, though both rely on refinements or synthetic pretraining.

In contrast, our Hyden framework integrates a full-resolution CNN with a low-resolution ViT and self-distillation, leveraging CNNs' linear scaling to produce sharp, efficient, and generalizable predictions for depth and normals at megapixel scales.

## 3 Approach

### 3.1 Hybrid Dual-Path Vision Encoder

Our architecture employs a hybrid dual-path encoder that combines a low-resolution Vision Transformer (ViT) with a full-resolution CNN to balance global context and fine-detail preservation (Figure 2). The ViT branch processes a uniformly downsampled input (up to $518 \times 518$) to capture long-range dependencies at constant cost, leveraging any pretrained backbone (e.g., DepthAnything Yang et al. (2024)).

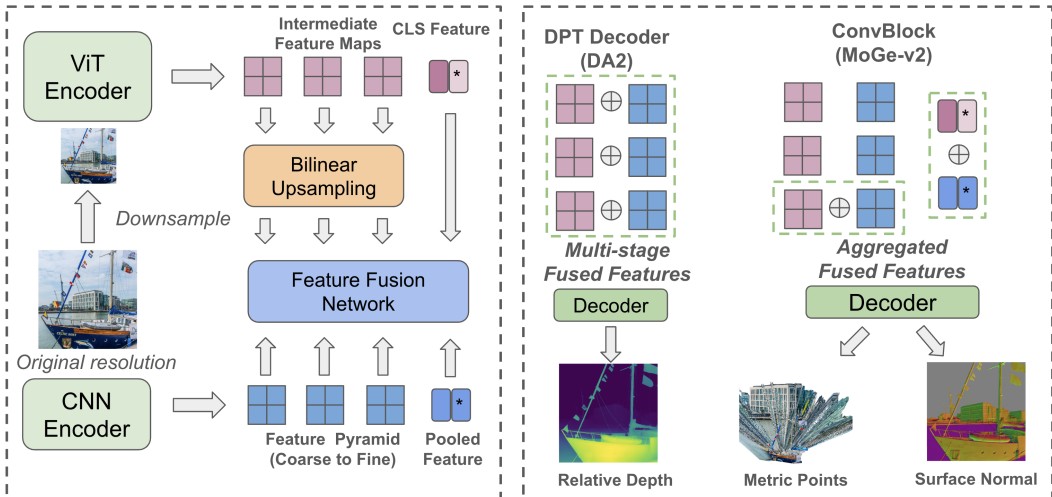

Figure 2: **Network Architecture:** The ViT encoder takes in down-sampled images while the CNN encoder takes in images with original resolution. To recover the high resolution features, the target ViT feature maps are upsampled with bilinear interpolation. CNN and ViT feature maps will be **concatenated** and fed into a feature fusion network. The fused features are used for down-stream tasks. Depending on the decoder architecture, the fusion logic needs to be slightly modified. For example, DA2 (DepthAnything-v2) uses all intermediate features from ViT and we fuse the corresponding CNN layers for each ViT features. MoGe2 uses an aggregated ViT feature map and similarly we aggregate multi-scale CNN feature maps with upsampling and concatenation, and only fuse the aggregated CNN and ViT features. For global-level feature, we apply average pooling to the CNN maps and concatenate the result with the CLS token for downstream tasks.

In parallel, the CNN branch directly processes the full-resolution image, efficiently extracting high-frequency features such as edges and textures. We adopt a ResNet-like encoder with hierarchical downsampling stages.

The two streams are fused by upsampling ViT features to the CNN resolution, concatenating them, and applying a lightweight two-layer convolution. This enables joint reasoning over global transformer context and local detail with minimal overhead.

A key advantage is scalability: ViT cost remains fixed while CNN scales linearly with resolution, enabling efficient inference on multi-megapixel images—unlike pure-ViT models with quadratic cost. The encoder is modular and task-agnostic, and we integrate it into DepthAnythingV2 Yang et al. (2024) and MoGe-V2 Wang et al. (2025) with minimal modifications to the fusion logic, preserving resolution robustness and efficiency (see supplemental materials).

## 3.2 Self-distillation Training

**Motivation.** High-resolution supervision is difficult to obtain in practice: real datasets rarely provide dense, clean depth or surface-normal labels at megapixel scales due to hardware and annotation constraints, while synthetic datasets introduce a domain gap relative to real imagery. To build a general training pipeline that upgrades an existing depth, point map or surface normal model to our hybrid dual-path encoder—and scales gracefully to high-resolution inputs—we introduce a *self-distillation* framework.

**Overview.** From a set of unlabeled high-resolution images $\{I\}$, we generate pseudo labels with a target model $\mathcal{T}$ (e.g., a strong zero-shot predictor). We extract (i) *global* labels from the down-sampled full image ($518 \times 518$), and (ii) *local* labels from $518 \times 518$ crops, which recover sharper details. Local labels may vary in scale and shift, so we align them to the global prediction before training (Figure 3).

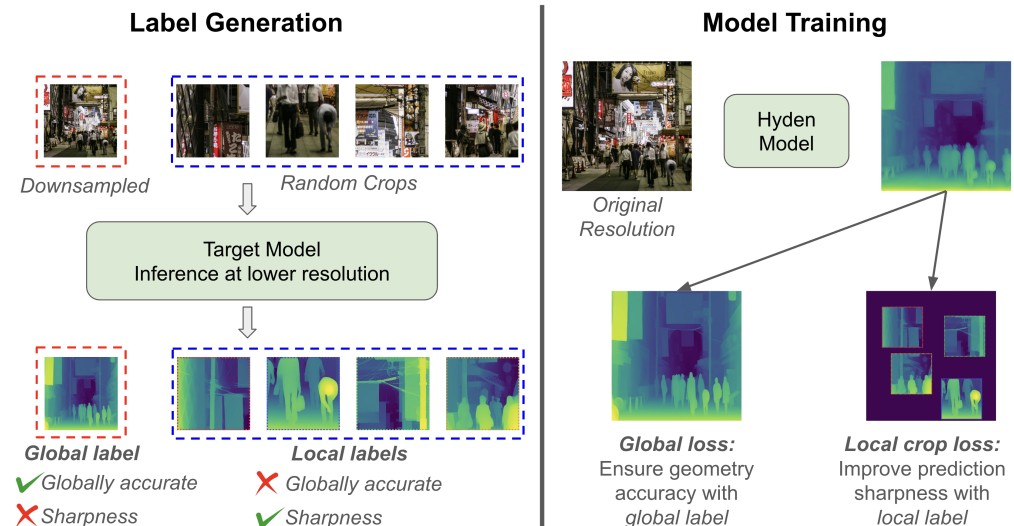

Figure 3: **Self-distillation:** (1) Label generation: Our pipeline samples multiple low-resolution views of the input, using downsampling for global context and random cropping for local details. The target model produces predictions on each view, which are mapped back to the original resolution via up-sampling or indexing. This yields pseudo labels that preserve geometric accuracy and local sharpness. (2) Model training: Using hyden encoders, models are trained on high-resolution inputs at their native resolution. A global loss is applied on downsampled predictions with global labels to retain geometry, while a local loss on full-resolution predictions enhances sharpness.

We then replace the target model's encoder with our Hyden encoder, adapt the decoder, and *freeze* the ViT branch—training only the CNN branch, fusion module, and decoder. Since the ViT sees only the downsampled view, it requires no additional fine-tuning.

**Notation.** Let $I \in \mathbb{R}^{H \times W \times 3}$ be a high-resolution image and $S$ denote the fixed low resolution ($S = 518$). Denote by $\downarrow_S (\cdot)$ uniform downsampling to $S \times S$, and by $\mathrm{crop}_k(\cdot)$ the $k$-th high-resolution crop operator with spatial support $\Omega_k \subseteq \{1, \ldots, H\} \times \{1, \ldots, W\}$; its resized version to $S \times S$ is $\mathrm{rcrop}_k(\cdot)$. The teacher $\mathcal{T}$ produces global pseudo labels $\mathbf{y}_g^{\mathrm{T}} = \mathcal{T}(\downarrow_S (I))$ and local pseudo labels $\mathbf{y}_k^{\mathrm{T}} = \mathcal{T}(\mathrm{rcrop}_k(I))$. Our student (hybrid) network $\mathcal{F}_\theta$ outputs a dense prediction $\mathbf{y} = \mathcal{F}_\theta(I)$ at the native resolution.

### 3.2.1 TASK-SPECIFIC LOSSES

We use the original task objectives of the target models for both global and local supervision.

**Relative depth (scale/shift-invariant).** Given predicted depth $d$ and teacher depth $\tilde{d}$ on a pixel set $\mathcal{M}$, we align the prediction by scale and shift

$$a^\star, b^\star = \arg\min_{a,b} \ \frac{1}{|\mathcal{M}|} \sum_{p \in \mathcal{M}} \left( a\, d_p + b - \tilde{d}_p \right)^2,$$

and compute a robust alignment loss (e.g., $\ell_1$):

$$\ell_{\mathrm{depth}}(d, \tilde{d}; \mathcal{M}) = \frac{1}{|\mathcal{M}|} \sum_{p \in \mathcal{M}} \left| a^\star d_p + b^\star - \tilde{d}_p \right| \tag{1}$$

**Surface normals (angular/cosine).** For predicted unit normals $n_p$ and teacher normals $\tilde{n}_p$,

$$\ell_{\mathrm{normal}}(n, \tilde{n}; \mathcal{M}) = \frac{1}{|\mathcal{M}|} \sum_{p \in \mathcal{M}} \left( 1 - \langle n_p, \tilde{n}_p \rangle \right) \tag{2}$$

**Affine-invariant point map loss.** Let $\hat{\mathbf{P}}_p^{\mathrm{aff}} \in \mathbb{R}^3$ be the predicted affine-invariant point map and $\tilde{\mathbf{P}}_p^{\mathrm{aff}}$ the teacher/GT affine-invariant point map for pixel $p \in \mathcal{M}$. The affine-invariant point map loss is:

$$\ell_{\mathrm{point}}^{\mathrm{aff}}(\hat{\mathbf{P}}^{\mathrm{aff}}, \tilde{\mathbf{P}}^{\mathrm{aff}}; \mathcal{M}) = \frac{1}{|\mathcal{M}|} \sum_{p \in \mathcal{M}} \left\| \hat{\mathbf{P}}_p^{\mathrm{aff}} - \tilde{\mathbf{P}}_p^{\mathrm{aff}} \right\|_1 \tag{3}$$

**Scale prediction loss.** Given the predicted global scale $\hat{s} > 0$, we supervise it against the optimal alignment scale $s^\star$ between $\hat{\mathbf{P}}^{\mathrm{aff}}$ and the metric GT points $\mathbf{P}_p$:

$$\mathcal{L}_{\mathrm{scale}} = \|\log \hat{s} - \mathrm{stopgrad}(\log s^\star)\|_2^2 \tag{4}$$

Please refer to DA2 Yang et al. (2024) and MoGe2 Wang et al. (2025) for more details in task-specific loss formulation. We write $\ell_{\mathrm{task}}$ to denote either $\ell_{\mathrm{depth}}$ or $\ell_{\mathrm{normal}}$ depending on the task.

### 3.2.2 GLOBAL LOSS

We downsample the student prediction to $S \times S$ and compare it with the global pseudo label:

$$\mathcal{L}_{\mathrm{global}} = \ell_{\mathrm{task}}\Big( \downarrow_S (\mathbf{y}), \, \mathbf{y}_{\mathrm{g}}^{\mathrm{T}}; \, \mathcal{M}_{\mathrm{g}} \Big) \tag{5}$$

where $\mathcal{M}_{\mathrm{g}}$ is the valid-pixel mask at the global resolution (e.g., invalid depth/normal entries removed).

### 3.2.3 LOCAL CROP LOSS

For each crop $k$, we compare the high-resolution student prediction with the teacher labels projected back to the crop region. Let $\uparrow_{\Omega_k}(\cdot)$ denote injecting local labels at $S \times S$ to the high-resolution support $\Omega_k$, and $M_k$ be the binary mask of $\Omega_k$:

$$\mathcal{L}_{\mathrm{local}} = \frac{1}{K} \sum_{k=1}^{K} \ell_{\mathrm{task}}\Big(\mathbf{y}, \, \uparrow_{\Omega_k}\big(\mathbf{y}_k^{\mathrm{T}}\big); \, \mathcal{M}_k\Big),$$
$$\text{where} \quad \mathcal{M}_k = M_k \cap \mathrm{valid}\big(\uparrow_{\Omega_k}\big(\mathbf{y}_k^{\mathrm{T}}\big)\big). \tag{6}$$

This formulation applies the task loss *directly at the native image resolution* but only within each crop's support. The loss is averaged across all crops.

### 3.2.4 TOTAL OBJECTIVE

The final objective combines global and local terms:

$$\mathcal{L}_{\mathrm{total}} = \lambda_{\mathrm{g}} \, \mathcal{L}_{\mathrm{global}} + \lambda_\ell \, \mathcal{L}_{\mathrm{local}},$$

with weights $\lambda_{\mathrm{g}}, \lambda_\ell = 1$. In all experiments we freeze the ViT branch and optimize the CNN encoder, fusion layer, and task decoder end-to-end using $\mathcal{L}_{\mathrm{total}}$.

## 3.3 IMPLEMENTATION DETAILS

We adapt our Hyden architecture to DA2 and MoGe2, denoted as Hyden-DA2 and Hyden-MoGe2. The additional CNN encoder introduces 10M parameters, incurring only a minor computational overhead. During self distillation, we randomly resize the input images from 518 to 2072 resolution for better geometry consistency (see Section 4.2). We use 4 local crops for all our self-distillation experiments. For the unlabeled high-resolution images $\{I\}$, we sampled 50 million images from a publicly available repository of crawled web data and we resized all the images to 2072x2072 resolution. We train our models for 300k iterations with batch size 192 on 64 NVIDIA H100 GPUs. We use an initial learning rate of 1e-5 for CNN encoder and 1e-6 for feature fusion and decoder module. We use adamW Kinga et al. (2015) optimizer and use polynomial learning rate scheduler.

## 4 RESULTS

### 4.1 BASELINE AND EVALUATION METRICS

We benchmark Hyden models against DepthAnythingV2 Yang et al. (2024) for relative depth and MoGe2 Wang et al. (2025) for metric-scale pointmaps and normals. We also compare with high-resolution depth methods DepthPro Bochkovskii et al. (2024) and FlashDepth Chou et al. (2025), and with normals-focused DSINE Bae & Davison (2024) and Metric3Dv2 Hu et al. (2024).

**Evaluation Metrics** For relative depth and metric pointmaps, we evaluate on 9 datasets: NYUv2 Nathan Silberman & Fergus (2012), KITTI Geiger et al. (2013), ETH3D Schops et al. (2017), iBims-1 Koch et al. (2018; 2020), DDAD Guizilini et al. (2020), DIODE Vasiljevic et al. (2019), HAMMER Jung et al. (2023), Booster Ramirez et al. (2022), and Middlebury Scharstein et al. (2014). These span indoor, street-view, and object domains, with ETH3D, Booster, and Middlebury providing 2K+ ground truth for high-resolution evaluation. We report the average relative error for point maps and depth:

$$\mathrm{Rel}_p = \frac{\|\hat{\mathbf{p}} - \mathbf{p}\|_2}{\|\mathbf{p}\|_2}, \quad \mathrm{Rel}_d = \frac{|\hat{z} - z|}{z},$$

along with the percentage of inliers

$$\delta_1^p : \frac{\|\hat{\mathbf{p}} - \mathbf{p}\|_2}{\|\mathbf{p}\|_2} < 0.25, \quad \delta_1^d : \max\left(\frac{\hat{d}}{d}, \frac{d}{\hat{d}}\right) < 1.25.$$

For surface normal, we evaluate on NYUv2 Nathan Silberman & Fergus (2012), iBims-1 Koch et al. (2018; 2020), Scannet Dai et al. (2017), and vkitti Cabon et al. (2020), reporting mean angular error. For boundary sharpness, we follow MoGe2 Wang et al. (2025) and evaluate on iBims-1, Sintel Butler et al. (2012), HAMMER, and Spring Mehl et al. (2023).

For all evaluation benchmarks, we resize the largest side of the input images to target image resolution and the predictions are evaluated at the original groundtruth resolution.

Table 1: **Zero-shot depth & point map accuracy.** We report the average relative error (lower is better) and $\delta_1$ score per dataset (higher is better) and aggregate performance across datasets via the average rank (lower is better). *DepthPro is evaluated at 1536x1536 and all other models are evaluated with 2K resolution input.

| Depth Model | Inference Latency (ms) | NYUv2 Rel↓ | δ₁↑ | KITTI Rel↓ | δ₁↑ | ETH3D Rel↓ | δ₁↑ | iBims-1 Rel↓ | δ₁↑ | Booster Rel↓ | δ₁↑ | Middlebury Rel↓ | δ₁↑ | DDAD Rel↓ | δ₁↑ | DIODE Rel↓ | δ₁↑ | HAMMER Rel↓ | δ₁↑ | Avg. Rank↓ |
|---|---|---|---|---|---|---|---|---|---|---|---|---|---|---|---|---|---|---|---|---|
| **Relative depth map** | | | | | | | | | | | | | | | | | | | | |
| DA2 Yang et al. (2024) | 408.1 | 5.4 | 92.3 | 8.3 | 92.3 | 5.5 | 94.3 | 4.1 | 95.5 | 3.0 | 98.8 | 7.3 | 87.8 | 15.8 | 82.4 | 5.4 | 95.8 | 6.3 | 96.1 | 4.6 |
| DepthPro Bochkovskii et al. (2024) | 341.3* | 4.4 | 96.5 | 5.7 | 95.8 | 7.5 | 93.1 | 4.2 | 96.7 | 3.2 | 98.6 | 9.2 | 84.8 | 15.1 | 80.1 | 4.9 | 94.3 | 5.3 | 98.3 | 4.3 |
| FlashDepth Chou et al. (2025) | **98.9** | 8.8 | 90.2 | 12.0 | 91.4 | 8.7 | 91.2 | 8.3 | 87.3 | 5.5 | 95.1 | 11.0 | 78.9 | 19.7 | 75.5 | 7.9 | 90.4 | 7.3 | 87.1 | 6.7 |
| Metric3Dv2 Hu et al. (2024) | 606.7 | 5.8 | 92.1 | 5.6 | 95.7 | 5.6 | 94.6 | 5.0 | 93.1 | 3.4 | 98.8 | 12.6 | 75.8 | **10.8** | **92.7** | **3.4** | **97.9** | 3.7 | 98.5 | 4.0 |
| MoGe2 Wang et al. (2025) | 476.8 | 3.9 | 97.3 | 5.0 | 96.9 | 3.8 | 98.1 | 3.3 | 98.2 | **2.1** | **99.2** | 2.3 | 94.3 | 11.3 | 90.5 | 3.9 | 96.7 | 3.4 | 98.8 | 2.0 |
| Hyden-DA2 (Ours) | **100.7** | 4.6 | 96.5 | 7.6 | 95.3 | 5.1 | 95.8 | 4.1 | 97.8 | 3.0 | 98.7 | 10.4 | 83.2 | 13.4 | 85.7 | 4.7 | 96.2 | 5.4 | 97.3 | 3.9 |
| Hyden-MoGe2 (Ours) | 171.6 | **3.7** | **98.5** | **4.9** | **97.8** | 3.8 | **98.3** | **3.2** | 98.6 | 2.1 | 99.0 | **2.0** | **95.8** | 11.1 | 91.6 | 3.7 | 97.1 | **3.2** | **99.1** | 1.3 |
| **Metric depth map (w/o GT intrinsics)** | | | | | | | | | | | | | | | | | | | | |
| DepthPro Bochkovskii et al. (2024) | 341.3* | 11.7 | 89.7 | 25.8 | 34.2 | 38.1 | 31.9 | 16.4 | 79.4 | 45.3 | 38.4 | - | - | 35.1 | 36.4 | 32.7 | 35.6 | 39.0 | 61.7 | 3.0 |
| MoGe2 Wang et al. (2025) | 476.8 | 9.2 | 92.5 | **15.7** | **87.1** | 19.6 | 82.3 | 14.8 | 88.6 | 24.9 | 35.6 | - | - | 25.3 | 55.6 | 18.0 | 69.4 | **24.1** | **68.3** | 1.8 |
| Hyden-MoGe2 (Ours) | **171.6** | **7.7** | **96.1** | 16.4 | 86.2 | **18.3** | **85.7** | **12.4** | **91.2** | **20.2** | 47.3 | - | - | **24.1** | 59.2 | 17.7 | **71.4** | 26.0 | 64.6 | 1.3 |
| **Metric point map** | | | | | | | | | | | | | | | | | | | | |
| DepthPro Bochkovskii et al. (2024) | 341.3* | 12.1 | 88.3 | 26.1 | 67.8 | 40.2 | 61.9 | 18.9 | 75.6 | 72.8 | 32.8 | - | - | 36.7 | 34.4 | 32.1 | 34.9 | 40.3 | 58.9 | 3.0 |
| MoGe2 Wang et al. (2025) | 476.8 | 9.7 | 93.8 | **16.8** | **85.7** | 20.3 | 92.6 | 16.2 | 84.6 | 63.3 | 38.3 | - | - | 26.5 | 53.4 | 18.0 | 69.6 | **25.2** | **69.5** | 1.8 |
| Hyden-MoGe2 (Ours) | **171.6** | **8.3** | **95.7** | 17.9 | 83.1 | **19.1** | **94.7** | **14.1** | **90.7** | **50.8** | 49.3 | - | - | **25.4** | 57.9 | 17.7 | **72.9** | 28.7 | 66.3 | 1.3 |

### 4.2 PERFORMANCE COMPARISON ON IMAGES WITH DIFFERENT RESOLUTIONS

We first evaluate robustness of test-time input scaling. As illustrated in Figure 1, at low resolution (518×518), DA2 Yang et al. (2024) and MoGe2 Wang et al. (2025) surpass Hyden, but above 784×784 all baselines, including Metric3Dv2 Hu et al. (2024), degrade sharply while Hyden stays accurate. Although FlashDepth Chou et al. (2025) offers relatively low inference latency,

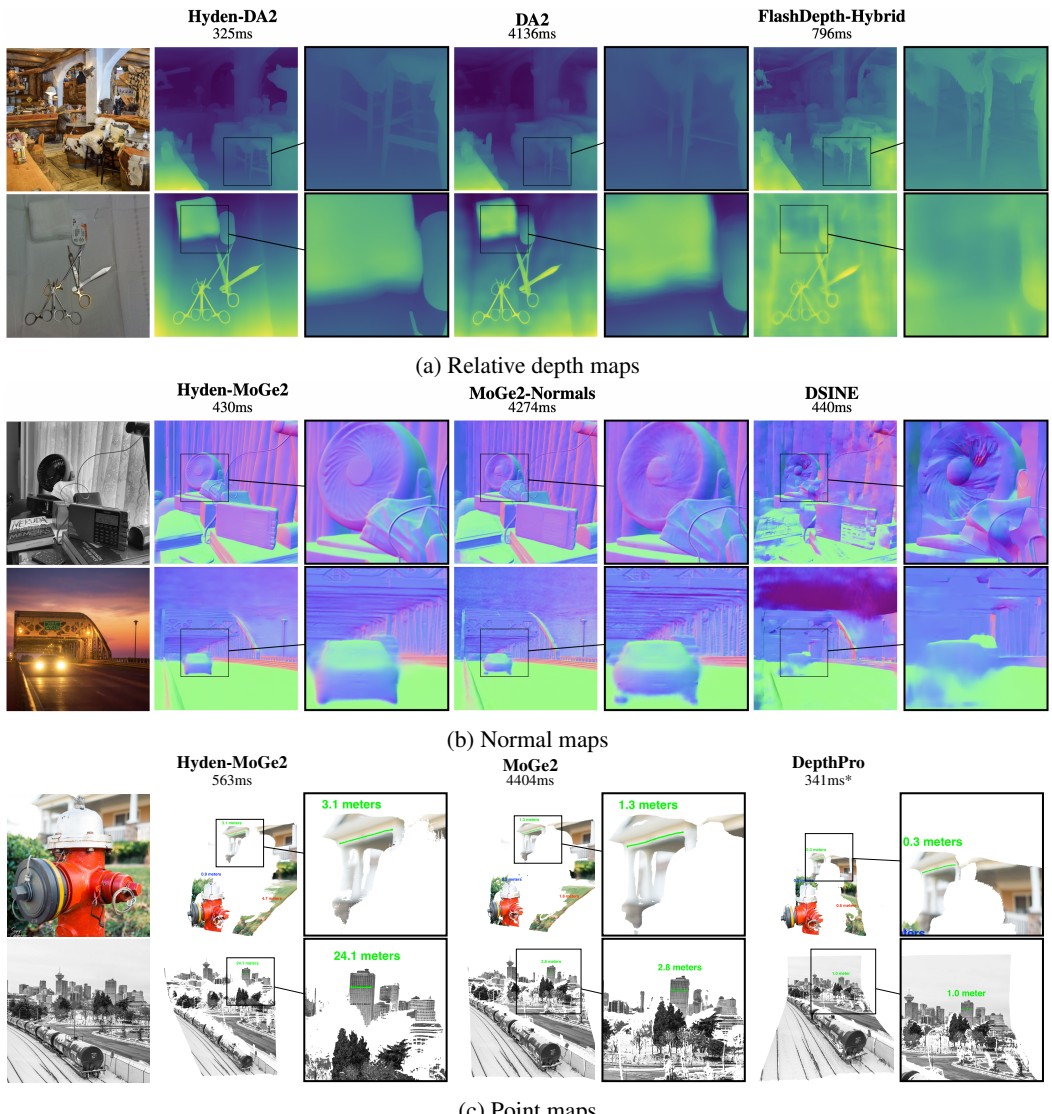

Figure 4: Qualitative comparison of geometry predictions on in-the-wild images at $4004 \times 4004$. Left: Hyden models; Middle: base models (DepthAnythingV2, MoGe2); Right: other SOTA models. Hyden models produces sharper, more accurate geometry with much lower latency. (*DepthPro is evaluated at 1536x1536 resolution due to model constraints.)

its lightweight decoder and limited supervision result in significant performance drops. Hyden models maintain consistent depth and pointmap accuracy across resolutions, with only marginal latency increases. Trained on mixed resolutions (Section 5.2), it combines stability with efficient high-res inference. In particular, Hyden-MoGe2 delivers the best 2K+ accuracy and low latency, and at 4K (Figure 1) outperforms ViT baselines while running 10× faster.

### 4.3 PERFORMANCE COMPARISON ON HIGH-RESOLUTION IMAGES

**Zero-shot depth & point map**   As shown in Figure 4, Hyden-DA2 yields sharper geometry and Hyden-MoGe2 predict better metric scale at high resolution. In Table 1, Hyden-MoGe2 attains the highest average accuracy, while both Hyden variants improve accuracy and reduce latency over their baselines. At 2K, Hyden-MoGe2 achieves the best geometric accuracy and runs ∼3× faster, highlighting the effectiveness of our self-distillation for high-resolution accuracy and efficiency.

**Zero-shot surface normal**  As shown in Table 2, Hyden-MoGe2 improves accuracy on most benchmarks while keeping low latency, with qualitative results in Figure 4. DSINE Bae & Davison (2024) runs at similar speed but trails ViT-based models. In out-of-domain tests, Hyden-MoGe2 surpasses Metric3Dv2 on three benchmarks and significantly outperforms MoGe2 at 2K resolution.

Table 2: **Zero-shot surface normal accuracy.** We report the mean angular errors (lower is better). *ScanNet evaluation is in-domain for Metric3Dv2, as it is included in the training set.

| Surface normal
Model | Inference
Latency (ms) | NYUv2
Mean↓ | iBims-1
Mean↓ | Scannet
Mean↓ | vkitti
Mean↓ | Avg.
Rank↓ |
|---|---|---|---|---|---|---|
| DSINE Bae & Davison (2024) | 149.4 | 17.1 | 18.0 | 16.9 | 30.2 | 4.0 |
| Metric3Dv2 Hu et al. (2024) | 606.7 | 15.9 | 15.4 | **11.4*** | 29.6 | 2.3 |
| MoGe2 Wang et al. (2025) | 438.2 | 15.6 | 16.0 | 13.7 | 27.3 | 2.5 |
| Hyden-MoGe2 (Ours) | **127.4** | **14.6** | **14.8** | 13.0 | **27.0** | **1.2** |

Table 3: **Zero-shot boundary sharpness.** We report F1 score and recall for all datasets (higher is better).

| Depth
Model | iBims-1
F1↑ | R↑ | Sintel
F1↑ | R↑ | HAMMER
F1↑ | R↑ | Spring
F1↑ | R↑ | Avg.
Rank↓ |
|---|---|---|---|---|---|---|---|---|---|
| DA2 Yang et al. (2024) | 12.7 | 20.0 | 28.7 | 36.4 | 7.7 | 13.4 | 16.3 | 15.3 | 5.0 |
| FlashDepth Chou et al. (2025) | 11.3 | 11.2 | 25.3 | 28.7 | 6.0 | 5.9 | 20.2 | 17.6 | 5.7 |
| Metric3Dv2 Hu et al. (2024) | 12.8 | 13.4 | 22.7 | 24.4 | 4.9 | 4.0 | 17.6 | 14.1 | 6.0 |
| DepthPro Bochkovskii et al. (2024) | 49.2 | 43.1 | 40.3 | 44.1 | 7.5 | 7.3 | 37.1 | 33.9 | 2.2 |
| MoGe2 Wang et al. (2025) | 49.0 | 45.6 | 38.2 | 41.4 | 7.4 | 7.2 | 34.8 | 32.5 | 3.3 |
| Hyden-DA2 (Ours) | 15.8 | 21.3 | 33.1 | 46.0 | **10.7** | **19.3** | 15.9 | 16.8 | 4.0 |
| Hyden-MoGe2 (Ours) | **54.7** | **50.4** | **46.5** | **49.6** | 7.9 | 7.6 | **34.2** | **29.9** | 1.6 |

**Zero-shot boundary sharpness**  As shown in Table 3, Hyden models achieve notably higher F1 and recall than their baselines while running faster at 2K. Hyden-MoGe2 further surpasses other SOTA methods without extra high-resolution supervision.

Additional results and model size comparisons are provided in the supplemental material.

## 5  ABLATION STUDY

### 5.1  IMPORTANCE OF LOCAL CROP LOSS

As shown in Table 4 (all models are evaluated with 2K resolution input), removing the local crop loss causes Hyden-DA2 to lose sharpness due to reliance on low-resolution global supervision. Increasing the number of crops improves sharpness, and we find that using four crops offers best trade-off between labeling cost and model performance.

Table 4: **Ablation on local crop loss.** (We report F1 score and recall for zero-shot boundary sharpness evaluation.)

| Depth
Model | iBims-1
F1↑ | R↑ | Sintel
F1↑ | R↑ | HAMMER
F1↑ | R↑ | Spring
F1↑ | R↑ |
|---|---|---|---|---|---|---|---|---|
| Hyden-DA2 w/o local crop loss | 11.8 | 18.4 | 27.9 | 38.2 | 7.8 | 13.1 | 14.7 | 13.8 |
| Hyden-DA2 w/ 2 crops | 14.4 | 20.9 | 31.8 | 40.5 | 8.7 | 16.8 | 15.5 | 14.7 |
| Hyden-DA2 w/ 4 crops | 15.8 | 21.3 | **33.1** | **46.0** | **10.7** | **19.3** | 15.9 | 16.8 |
| Hyden-DA2 w/ 8 crops | **16.1** | **22.2** | 32.3 | 45.7 | 10.3 | 18.9 | **17.1** | **18.5** |

Table 5: **Ablation on mixed-resolution training & fusion network design.**

| Depth
Model | NYUv2
Rel↓ | KITTI
Rel↓ | ETH3D
Rel↓ | HAMMER
Rel↓ |
|---|---|---|---|---|
| Hyden-DA2 trained from 518-1036 | 5.14 | 8.83 | 5.27 | 7.10 |
| Hyden-DA2 trained from 518-2072 | **4.60** | **7.63** | **5.12** | **5.44** |
| Hyden-DA2 w/ MLP Fusion | 4.72 | 7.92 | 5.31 | 5.93 |
| Hyden-DA2 w/ 1-layer CNN Fusion | 4.63 | 7.88 | 5.22 | 5.87 |
| Hyden-DA2 w/ 2-layer CNN Fusion | **4.60** | **7.63** | **5.12** | **5.44** |

### 5.2  IMPORTANCE OF MIXED-RESOLUTION TRAINING

Table 5 compares Hyden models trained with input resolutions ranging from 518–1036 and 518–2072. Matching the training resolution to the test-time resolution improves depth accuracy, highlighting the value of mixed-resolution training. However, for ViT encoders, training at very high resolutions (e.g., over 20K tokens at 2K resolution) is computationally prohibitive. By constraining the ViT branch to low-resolution input, Hyden enables practical high-resolution training.

### 5.3  FEATURE FUSION NETWORK DESIGN

We evaluate several fusion designs for feature projection: a single linear layer, a single CNN layer, and two CNN layers with ReLU activation. As shown in Table 5, the two-layer CNN achieves the best performance. We adopt this configuration in our model as it offers superior accuracy without incurring excessive computational overhead.

## 6 CONCLUSION

We presented Hyden, a hybrid dual-path vision encoder that delivers high-resolution depth, point map and surface-normal estimation with low latency. By combining global and local pseudo-label self-distillation, Hyden preserves geometric accuracy while enhancing fine details, without relying on high-resolution ground truth. Integrated into leading baselines, Hyden achieves state-of-the-art accuracy across resolutions while maintaining fast inference, offering a scalable solution for dense prediction tasks.

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
