# HYDEN: A HYBRID DUAL-PATH ENCODER FOR MONOCULAR GEOMETRY OF HIGH-RESOLUTION IMAGES SUPPLEMENTAL MATERIAL

**Zaiwei Zhang & Marc Mapeke & Wei Ye & Rakesh Ranjan & JQ Huang**
Meta Reality Labs
322 Airport Blvd.
Burlingame, CA 94010, USA
{zaiweizhang,mmapeke,weiye,rakeshr,jqhuang}@meta.com

## A   CNN MODEL ARCHITECTURE

Motivated from MobileNets Howard et al. (2017), we use an efficient CNN architecture, where most of the CNN blocks consume depth-wise convolution combined with linear layers. As shown from Figure 1, we also use ResNet-style skip links within each CNN block. The CNN backbone generates a pyramid of features for feature fusion in the later stage. The CNN network has 10M parameters and is pre-trained with web-crawled data.

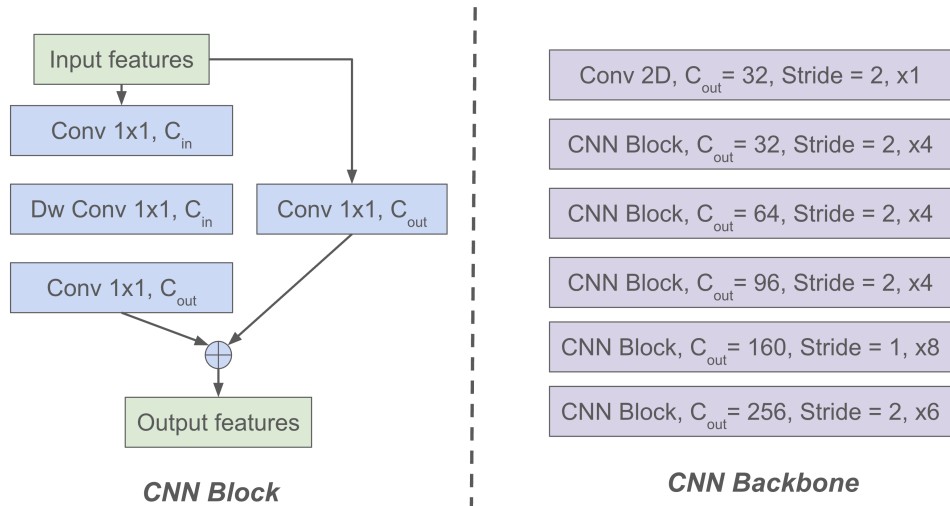

Figure 1: Left: CNN block layer design. Right: CNN backbone design.

## B   FUSION LAYER DETAILS

In DepthAnythingV2 Yang et al. (2024), the decoder takes a ViT feature pyramid and then decodes feature maps from coarse to fine. As shown from Figure 2, we concatenate similar feature maps from CNN feature pyramid with projected ViT feature maps. To match the size, we will up-sample the ViT features with bilinear interpolation. We match all the features from coarse to fine.

In MoGe-2 Wang et al. (2025), the decoder takes in all the features from ViT and aggregate them into one single feature map. Similarly, we also aggregate all the CNN features with feature concatenation. To match the size, we up-sample the low-resolution CNN feature map with bilinear interpolation. All the CNN feature maps are up-sampled to the highest CNN feature map resolution. Since the number of tokens do not change during feature propagation, ViT feature maps have the same size. We also up-sample the aggregated ViT feature map and then concatenate with the aggregated CNN features as shown from Figure 3.

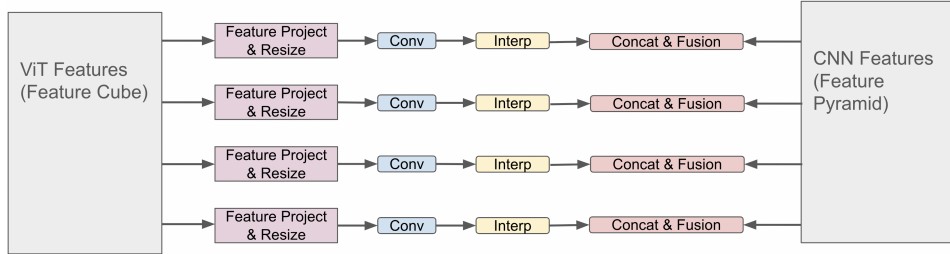

Figure 2: Feature fusion for Hyden-DA2

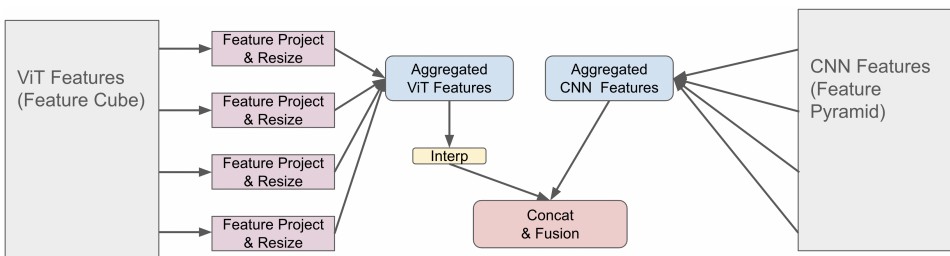

Figure 3: Feature fusion for Hyden-MoGe2

## C  MODEL SIZE AND INFERENCE LATENCY COMPARISON

Table 1: Model inference latency across different input resolutions. The metric is milisecond (the lower the better). Due to model constraint, we only evalaute DepthPro at 1536x1536 resolution.

| Resolution | Hyden-DA2 | DA2 | FlashDepth | Hyden-MoGe2 | MoGe2 | DepthPro | Metric3Dv2 |
|---|---|---|---|---|---|---|---|
| 518 | 35.92 | 21.91 | 33.26 | 46.86 | 35.08 | - | 65.38 |
| 784 | 40.51 | 35.86 | 38.34 | 54.59 | 51.55 | - | 80.28 |
| 1036 | 44.72 | 60.69 | 35.63 | 63.78 | 82.42 | - | 120.44 |
| 1554 | 67.70 | 170.42 | 54.55 | 106.36 | 211.79 | 341.37 | 284.12 |
| 2072 | 100.74 | 408.06 | 98.85 | 171.60 | 476.79 | - | 606.72 |
| 2590 | 146.03 | 860.02 | 181.59 | 248.77 | 962.88 | - | 1170.57 |
| 3108 | 205.37 | 1644.76 | 328.80 | 345.42 | 1793.50 | - | 2106.68 |
| 3584 | 256.67 | 2749.84 | 535.77 | 450.07 | 2946.23 | - | 3369.72 |
| 4004 | 325.14 | 4136.93 | 795.63 | 563.06 | 4404.79 | - | 4922.21 |
| model_size | 400M | 334M | 360M | 374M | 326M | 952M | 412M |

We compare the model sizes and also the inference latency across different resolution in Table 1. As shown in the table, models with hyden encoders are gaining more advantages with higher input image resolution. Especially, at above 2K resolution, hyden variants are significantly faster compared to other SOTA methods.

## D  MORE QUANTITATIVE RESULTS

From Table 2 to Table 10, we show the relative depth evaluation across different resolutions for multiple datasets. We can observe that as input image resolution gets higher, SOTA methods (such as DA2, FlashDepth, MoGe2 and Metric3Dv2) all experience significant performance regression (especially in NYU, iBims-1, DIODE and HAMMER). However, our hyden variants remain accurate for most datasets. We also show surface normal, metric depth and metric point map evaluation across different resolutions for selected datasets in Table 13, Table 11 and Table 12. We observe similar trend for all three different tasks. From Table 16 to Table 17, we show the sharpness evaluation with relative depth across different resolutions. We observe that with higher image resolution, the F1 scores get higher for all datasets. Our hyden variants still generate the highest F1 scores across different datasets among other SOTA methods.

Table 2: Relative depth evaluation on NYU across resolutions. The metric is Abs-rel (the lower the better). Due to model constraint, we only evalaute DepthPro at 1536x1536 resolution.

| Resolution | Hyden-DA2 | DA2 | FlashDepth | Hyden-MoGe2 | MoGe2 | DepthPro | Metric3Dv2 |
|---|---|---|---|---|---|---|---|
| 518 | 4.5 | 4.7 | 7.0 | 3.7 | 3.8 | - | 5.6 |
| 784 | 4.6 | 4.7 | 6.9 | 3.7 | 3.7 | - | 5.6 |
| 1036 | 4.6 | 4.8 | 7.0 | 3.7 | 3.7 | - | 4.9 |
| 1554 | 4.7 | 5.2 | 7.8 | 3.8 | 3.8 | 4.4 | 5.4 |
| 2072 | 4.6 | 5.4 | 8.8 | 3.7 | 3.9 | - | 5.8 |
| 2590 | 4.9 | 5.7 | 9.6 | 3.7 | 4.1 | - | 6.0 |
| 3108 | 4.7 | 5.9 | 10.6 | 3.4 | 4.3 | - | 6.2 |
| 3584 | 4.7 | 6.0 | 11.2 | 3.4 | 4.5 | - | 6.5 |
| 4004 | 4.7 | 6.2 | 11.8 | 3.8 | 4.6 | - | 7.1 |

Table 3: Relative depth evaluation on KITTI across resolutions. The metric is Abs-rel (the lower the better). Due to model constraint, we only evalaute DepthPro at 1536x1536 resolution.

| Resolution | Hyden-DA2 | DA2 | FlashDepth | Hyden-MoGe2 | MoGe2 | DepthPro | Metric3Dv2 |
|---|---|---|---|---|---|---|---|
| 518 | 7.7 | 8.6 | 13.8 | 5.3 | 5.8 | - | 5.3 |
| 784 | 7.7 | 8.7 | 15.2 | 5.1 | 5.4 | - | 5.3 |
| 1036 | 7.6 | 8.0 | 14.3 | 5.5 | 5.1 | - | 5.4 |
| 1554 | 7.4 | 7.6 | 12.4 | 5.1 | 4.9 | 5.7 | 5.8 |
| 2072 | 7.6 | 8.3 | 12.0 | 4.9 | 5.0 | - | 5.6 |
| 2590 | 7.7 | 8.7 | 11.3 | 5.3 | 5.1 | - | 5.6 |
| 3108 | 7.6 | 9.0 | 11.9 | 5.1 | 5.2 | - | 5.8 |
| 3584 | 7.6 | 9.3 | 12.0 | 5.1 | 5.4 | - | 6.1 |
| 4004 | 7.7 | 9.2 | 12.1 | 5.1 | 5.6 | - | 6.4 |

Table 4: Relative depth evaluation on ETH3D across resolutions. The metric is Abs-rel (the lower the better). Due to model constraint, we only evalaute DepthPro at 1536x1536 resolution.

| Resolution | Hyden-DA2 | DA2 | FlashDepth | Hyden-MoGe2 | MoGe2 | DepthPro | Metric3Dv2 |
|---|---|---|---|---|---|---|---|
| 518 | 5.3 | 5.6 | 11.3 | 4.6 | 4.5 | - | 5.3 |
| 784 | 5.3 | 5.2 | 10.2 | 4.2 | 4.3 | - | 4.1 |
| 1036 | 5.4 | 5.4 | 9.1 | 3.8 | 4.2 | - | 3.9 |
| 1554 | 5.4 | 5.7 | 9.2 | 4.1 | 3.8 | 7.5 | 4.5 |
| 2072 | 5.1 | 5.5 | 8.7 | 3.8 | 3.8 | - | 5.6 |
| 2590 | 5.4 | 5.6 | 8.7 | 3.8 | 4.1 | - | 6.1 |
| 3108 | 5.4 | 5.7 | 9.9 | 4.2 | 4.2 | - | 6.6 |
| 3584 | 5.3 | 5.9 | 9.5 | 3.9 | 4.3 | - | 6.9 |
| 4004 | 5.2 | 6.0 | 9.6 | 4.0 | 5.0 | - | 7.2 |

Table 5: Relative depth evaluation on iBims-1 across resolutions. The metric is Abs-rel (the lower the better). Due to model constraint, we only evalaute DepthPro at 1536x1536 resolution.

| Resolution | Hyden-DA2 | DA2 | FlashDepth | Hyden-MoGe2 | MoGe2 | DepthPro | Metric3Dv2 |
|---|---|---|---|---|---|---|---|
| 518 | 4.3 | 4.2 | 8.3 | 3.5 | 3.3 | - | 4.8 |
| 784 | 4.0 | 4.5 | 7.9 | 3.3 | 3.1 | - | 4.4 |
| 1036 | 4.0 | 4.6 | 7.6 | 3.2 | 3.1 | - | 4.4 |
| 1554 | 4.1 | 4.4 | 7.6 | 3.2 | 3.2 | 4.2 | 4.8 |
| 2072 | 4.1 | 4.1 | 8.3 | 3.2 | 3.3 | - | 5.0 |
| 2590 | 4.1 | 4.4 | 8.7 | 3.2 | 3.3 | - | 5.1 |
| 3108 | 4.1 | 4.6 | 9.1 | 3.3 | 3.4 | - | 5.4 |
| 3584 | 4.2 | 5.0 | 9.5 | 3.3 | 3.6 | - | 5.7 |
| 4004 | 4.2 | 5.1 | 9.9 | 3.3 | 3.6 | - | 6.0 |

Table 6: Relative depth evaluation on Booster across resolutions. The metric is Abs-rel (the lower the better). Due to model constraint, we only evalaute DepthPro at 1536x1536 resolution.

| Resolution | Hyden-DA2 | DA2 | FlashDepth | Hyden-MoGe2 | MoGe2 | DepthPro | Metric3Dv2 |
|---|---|---|---|---|---|---|---|
| 518 | 2.9 | 2.7 | 5.1 | 2.1 | 2.1 | - | 3.1 |
| 784 | 2.9 | 2.7 | 5.0 | 2.0 | 2.0 | - | 2.9 |
| 1036 | 3.0 | 3.0 | 5.1 | 2.0 | 2.0 | - | 3.1 |
| 1554 | 3.0 | 3.2 | 5.1 | 2.1 | 2.0 | 3.2 | 3.4 |
| 2072 | 3.0 | 3.0 | 5.5 | 2.1 | 2.1 | - | 3.4 |
| 2590 | 3.0 | 3.5 | 5.7 | 2.1 | 2.1 | - | 3.4 |
| 3108 | 3.0 | 3.7 | 6.0 | 2.1 | 2.2 | - | 3.6 |
| 3584 | 3.0 | 3.8 | 6.2 | 2.1 | 2.2 | - | 3.6 |
| 4004 | 3.0 | 3.9 | 6.5 | 2.1 | 2.3 | - | 3.7 |

Table 7: Relative depth evaluation on DIODE across resolutions. The metric is Abs-rel (the lower the better). Due to model constraint, we only evalaute DepthPro at 1536x1536 resolution.

| Resolution | Hyden-DA2 | DA2 | FlashDepth | Hyden-MoGe2 | MoGe2 | DepthPro | Metric3Dv2 |
|---|---|---|---|---|---|---|---|
| 518 | 4.5 | 4.7 | 7.0 | 3.7 | 3.8 | - | 3.1 |
| 784 | 4.6 | 4.7 | 6.9 | 3.7 | 3.7 | - | 2.7 |
| 1036 | 4.6 | 4.8 | 7.0 | 3.7 | 3.7 | - | 2.8 |
| 1554 | 4.7 | 5.2 | 7.8 | 3.8 | 3.8 | 4.9 | 3.7 |
| 2072 | 4.7 | 5.4 | 7.9 | 3.7 | 3.9 | - | 3.4 |
| 2590 | 4.9 | 5.7 | 8.5 | 3.7 | 4.1 | - | 3.7 |
| 3108 | 4.7 | 5.9 | 9.3 | 3.4 | 4.3 | - | 4.1 |
| 3584 | 4.7 | 6.0 | 9.8 | 3.4 | 4.5 | - | 4.5 |
| 4004 | 4.7 | 6.2 | 10.1 | 3.8 | 4.6 | - | 5.0 |

Table 8: Relative depth evaluation on DDAD across resolutions. The metric is Abs-rel (the lower the better). Due to model constraint, we only evalaute DepthPro at 1536x1536 resolution.

| Resolution | Hyden-DA2 | DA2 | FlashDepth | Hyden-MoGe2 | MoGe2 | DepthPro | Metric3Dv2 |
|---|---|---|---|---|---|---|---|
| 518 | 14.5 | 16.0 | 23.7 | 16.5 | 16.9 | - | 12.5 |
| 784 | 14.7 | 15.5 | 21.7 | 12.2 | 12.1 | - | 8.9 |
| 1036 | 14.3 | 15.3 | 20.7 | 11.8 | 12.0 | - | 8.2 |
| 1554 | 13.6 | 14.7 | 19.7 | 11.5 | 11.1 | 15.1 | 9.0 |
| 2072 | 13.4 | 15.8 | 19.7 | 11.1 | 11.3 | - | 10.8 |
| 2590 | 13.7 | 16.2 | 20.5 | 11.6 | 11.6 | - | 12.1 |
| 3108 | 13.9 | 17.6 | 21.3 | 11.7 | 12.1 | - | 12.9 |
| 3584 | 13.6 | 18.5 | 21.8 | 11.6 | 12.0 | - | 13.2 |
| 4004 | 13.7 | 18.5 | 22.2 | 11.6 | 12.3 | - | 14.1 |

Table 9: Relative depth evaluation on HAMMER across resolutions. The metric is Abs-rel (the lower the better). Due to model constraint, we only evalaute DepthPro at 1536x1536 resolution.

| Resolution | Hyden-DA2 | DA2 | FlashDepth | Hyden-MoGe2 | MoGe2 | DepthPro | Metric3Dv2 |
|---|---|---|---|---|---|---|---|
| 518 | 5.6 | 5.7 | 7.2 | 3.4 | 3.4 | - | 4.8 |
| 784 | 5.4 | 5.3 | 7.1 | 3.1 | 3.0 | - | 3.9 |
| 1036 | 5.3 | 5.4 | 7.0 | 3.2 | 3.1 | - | 3.7 |
| 1554 | 5.4 | 5.9 | 6.7 | 3.2 | 3.2 | 5.3 | 3.6 |
| 2072 | 5.4 | 6.3 | 7.3 | 3.2 | 3.4 | - | 3.7 |
| 2590 | 5.5 | 6.7 | 7.8 | 3.2 | 3.7 | - | 4.0 |
| 3108 | 5.5 | 7.0 | 8.6 | 3.2 | 4.0 | - | 4.3 |
| 3584 | 5.5 | 7.1 | 9.2 | 3.2 | 4.2 | - | 4.5 |
| 4004 | 5.4 | 7.5 | 9.6 | 3.2 | 4.3 | - | 4.6 |

Table 10: Relative depth evaluation on Middlebury across resolutions. The metric is Abs-rel (the lower the better). Due to model constraint, we only evalaute DepthPro at 1536x1536 resolution.

| Resolution | Hyden-DA2 | DA2 | FlashDepth | Hyden-MoGe2 | MoGe2 | DepthPro | Metric3Dv2 |
|---|---|---|---|---|---|---|---|
| 518 | 11.3 | 10.9 | 12.6 | 2.2 | 2.3 | - | 16.5 |
| 784 | 11.5 | 11.4 | 12.5 | 2.1 | 2.1 | - | 14.9 |
| 1036 | 11.4 | 8.9 | 12.3 | 2.1 | 2.2 | - | 15.3 |
| 1554 | 10.5 | 10.1 | 10.6 | 2.1 | 2.3 | 9.2 | 13.9 |
| 2072 | 10.4 | 7.3 | 11.0 | 2.1 | 2.3 | - | 12.6 |
| 2590 | 7.4 | 7.3 | 9.6 | 2.1 | 2.3 | - | 12.6 |
| 3108 | 7.3 | 7.4 | 11.1 | 2.1 | 2.3 | - | 3.6 |
| 3584 | 7.3 | 4.2 | 9.0 | 2.1 | 2.2 | - | 3.6 |
| 4004 | 7.3 | 4.3 | 9.0 | 2.0 | 2.2 | - | 3.9 |

Table 11: Metric depth evaluation on NYU, DDAD and ETH3D across resolutions. The metric is Abs-rel (the lower the better). H-MoGe2 represents Hyden-MoGe2.

| Resolution | NYU | | | DDAD | | | ETH3D | | |
|---|---|---|---|---|---|---|---|---|---|
| | H-MoGe2 | DepthPro | MoGe2 | H-MoGe2 | DepthPro | MoGe2 | H-MoGe2 | DepthPro | MoGe2 |
| 518 | 7.4 | - | 8.4 | 32.7 | - | 43.6 | 27.3 | - | 29.6 |
| 784 | 7.5 | - | 8.1 | 24.3 | - | 29.1 | 20.0 | - | 24.8 |
| 1036 | 7.6 | - | 7.8 | 24.7 | - | 28.2 | 19.9 | - | 22.0 |
| 1554 | 7.7 | 11.7 | 17.9 | 24.2 | 35.1 | 24.8 | 20.3 | 38.1 | 20.1 |
| 2072 | 7.7 | - | 9.2 | 24.1 | - | 25.3 | 18.3 | - | 19.6 |
| 2590 | 7.8 | - | 12.9 | 24.6 | - | 27.2 | 18.8 | - | 20.3 |
| 3108 | 7.9 | - | 17.6 | 25.1 | - | 28.9 | 18.8 | - | 20.7 |
| 3584 | 8.0 | - | 21.5 | 24.8 | - | 29.1 | 18.3 | - | 21.2 |
| 4004 | 8.1 | - | 24.7 | 24.8 | - | 31.0 | 18.5 | - | 24.9 |

Table 12: Metric point map evaluation on NYU, DDAD and ETH3D across resolutions. The metric is Abs-rel (the lower the better). H-MoGe2 represents Hyden-MoGe2.

| Resolution | NYU | | | DDAD | | | ETH3D | | |
|---|---|---|---|---|---|---|---|---|---|
| | H-MoGe2 | DepthPro | MoGe2 | H-MoGe2 | DepthPro | MoGe2 | H-MoGe2 | DepthPro | MoGe2 |
| 518 | 8.1 | - | 8.9 | 31.3 | - | 44.8 | 29.7 | - | 30.8 |
| 784 | 8.2 | - | 8.6 | 25.5 | - | 28.2 | 20.5 | - | 25.7 |
| 1036 | 8.2 | - | 8.4 | 25.7 | - | 27.3 | 20.1 | - | 23.2 |
| 1554 | 8.2 | 12.1 | 8.5 | 25.4 | 36.7 | 27.1 | 19.5 | 40.2 | 21.7 |
| 2072 | 8.3 | - | 9.7 | 25.4 | - | 26.5 | 19.1 | - | 20.3 |
| 2590 | 8.4 | - | 13.1 | 25.7 | - | 27.8 | 18.7 | - | 20.9 |
| 3108 | 8.5 | - | 17.3 | 25.4 | - | 28.4 | 18.1 | - | 21.2 |
| 3584 | 8.6 | - | 20.8 | 25.8 | - | 30.1 | 18.3 | - | 22.4 |
| 4004 | 8.7 | - | 23.7 | 25.9 | - | 31.9 | 18.3 | - | 24.8 |

Table 13: Surface normal evaluation on NYU and iBims-1 across resolutions. The metric is mean angular errors (the lower the better).

| Resolution | Nyu | | | | iBims-1 | | | |
|---|---|---|---|---|---|---|---|---|
| | Hyden-MoGe2 | DSINE | Metric3Dv2 | MoGe2 | Hyden-MoGe2 | DSINE | Metric3Dv2 | MoGe2 |
| 518 | 14.4 | 17.1 | 14.2 | 14.7 | 16.0 | 17.2 | 15.1 | 15.3 |
| 784 | 14.5 | 17.3 | 14.0 | 14.8 | 15.3 | 17.6 | 14.5 | 14.8 |
| 1036 | 14.4 | 17.2 | 14.1 | 14.9 | 14.8 | 17.5 | 14.2 | 14.8 |
| 1554 | 14.6 | 17.5 | 15.2 | 15.0 | 14.6 | 17.8 | 14.9 | 14.9 |
| 2072 | 14.7 | 17.6 | 15.9 | 15.5 | 14.8 | 18.0 | 15.4 | 15.3 |
| 2590 | 14.9 | 17.8 | 16.1 | 15.7 | 15.3 | 18.2 | 15.8 | 15.6 |
| 3108 | 15.2 | 18.0 | 16.1 | 16.2 | 15.9 | 18.7 | 16.0 | 16.0 |
| 3584 | 15.6 | 18.1 | 16.5 | 16.5 | 16.5 | 18.9 | 16.7 | 16.6 |
| 4004 | 15.7 | 18.2 | 17.4 | 16.8 | 16.9 | 18.7 | 17.2 | 17.1 |

Table 14: Relative depth boundary sharpness evaluation on iBims-1 across resolutions. The metric is F1 score (the higher the better). Due to model constraint, we only evalaute DepthPro at 1536x1536 resolution.

| Resolution | Hyden-DA2 | DA2 | FlashDepth | Hyden-MoGe2 | MoGe2 | DepthPro | Metric3Dv2 |
|---|---|---|---|---|---|---|---|
| 518 | 13.3 | 12.8 | 8.0 | 50.4 | 47.4 | - | 12.3 |
| 784 | 15.1 | 13.1 | 9.5 | 52.3 | 47.7 | - | 12.9 |
| 1036 | 15.3 | 13.1 | 10.4 | 53.2 | 48.1 | - | 12.7 |
| 1554 | 15.8 | 13.3 | 11.9 | 55.2 | 48.6 | 49.2 | 11.7 |
| 2072 | 15.8 | 12.7 | 11.3 | 54.7 | 49.0 | - | 12.8 |
| 2590 | 15.8 | 12.9 | 11.3 | 54.5 | 49.2 | - | 13.5 |
| 3108 | 16.2 | 12.0 | 10.0 | 53.9 | 49.4 | - | 13.0 |
| 3584 | 15.9 | 12.0 | 9.7 | 53.0 | 49.8 | - | 13.1 |
| 4004 | 16.3 | 11.8 | 9.1 | 53.3 | 49.6 | - | 11.1 |

Table 15: Relative depth boundary sharpness evaluation on Sintel across resolutions. The metric is F1 score (the higher the better). Due to model constraint, we only evalaute DepthPro at 1536x1536 resolution.

| Resolution | Hyden-DA2 | DA2 | FlashDepth | Hyden-MoGe2 | MoGe2 | DepthPro | Metric3Dv2 |
|---|---|---|---|---|---|---|---|
| 518 | 14.9 | 17.4 | 13.0 | 21.6 | 24.3 | - | 19.3 |
| 784 | 17.8 | 22.4 | 18.0 | 28.7 | 30.0 | - | 19.3 |
| 1036 | 25.6 | 25.0 | 20.7 | 35.5 | 33.4 | - | 20.9 |
| 1554 | 31.9 | 28.7 | 25.6 | 43.5 | 36.7 | 40.3 | 20.0 |
| 2072 | 33.1 | 28.7 | 25.3 | 46.5 | 38.2 | - | 22.7 |
| 2590 | 33.4 | 29.0 | 26.3 | 49.4 | 38.9 | - | 23.0 |
| 3108 | 33.5 | 28.1 | 24.4 | 50.0 | 39.0 | - | 24.0 |
| 3584 | 35.1 | 29.9 | 26.0 | 50.3 | 43.6 | - | 24.4 |
| 4004 | 33.8 | 26.5 | 22.2 | 51.2 | 38.8 | - | 25.1 |

Table 16: Relative depth boundary sharpness evaluation on HAMMER across resolutions. The metric is F1 score (the higher the better). Due to model constraint, we only evalaute DepthPro at 1536x1536 resolution.

| Resolution | Hyden-DA2 | DA2 | FlashDepth | Hyden-MoGe2 | MoGe2 | DepthPro | Metric3Dv2 |
|---|---|---|---|---|---|---|---|
| 518 | 7.5 | 6.8 | 5.4 | 3.8 | 7.0 | - | 3.3 |
| 784 | 8.9 | 7.7 | 6.2 | 5.5 | 7.6 | - | 5.3 |
| 1036 | 9.4 | 8.0 | 6.7 | 6.1 | 7.3 | - | 4.4 |
| 1554 | 10.3 | 7.8 | 6.7 | 7.3 | 7.9 | 7.5 | 4.6 |
| 2072 | 10.7 | 7.7 | 6.0 | 7.9 | 7.4 | - | 4.9 |
| 2590 | 10.5 | 7.6 | 6.4 | 7.6 | 7.0 | - | 4.6 |
| 3108 | 10.5 | 7.4 | 4.9 | 8.1 | 7.9 | - | 4.4 |
| 3584 | 10.6 | 7.6 | 4.9 | 8.7 | 7.5 | - | 4.3 |
| 4004 | 10.4 | 7.2 | 3.8 | 8.9 | 9.2 | - | 4.4 |

Table 17: Relative depth boundary sharpness evaluation on Spring across resolutions. The metric is F1 score (the higher the better). Due to model constraint, we only evalaute DepthPro at 1536x1536 resolution.

| Resolution | Hyden-DA2 | DA2 | FlashDepth | Hyden-MoGe2 | MoGe2 | DepthPro | Metric3Dv2 |
|---|---|---|---|---|---|---|---|
| 518 | 6.4 | 9.2 | 9.0 | 29.1 | 29.7 | - | 11.3 |
| 784 | 12.1 | 12.3 | 12.4 | 31.1 | 32.3 | - | 14.2 |
| 1036 | 13.7 | 14.3 | 15.1 | 32.9 | 34.0 | - | 15.7 |
| 1554 | 15.2 | 16.2 | 19.2 | 33.6 | 35.0 | 37.1 | 17.4 |
| 2072 | 15.9 | 16.3 | 20.2 | 34.2 | 34.8 | - | 17.6 |
| 2590 | 15.9 | 16.1 | 21.2 | 35.1 | 35.0 | - | 16.6 |
| 3108 | 16.5 | 15.3 | 20.5 | 36.0 | 35.3 | - | 15.9 |
| 3584 | 16.6 | 14.5 | 19.8 | 36.3 | 35.5 | - | 15.2 |
| 4004 | 16.1 | 13.9 | 18.7 | 36.5 | 35.1 | - | 14.3 |

# E  MORE QUALITATIVE RESULTS

In Figure 4, we show inference results across different input resolutions for different Hyden models. We show more qualitative comparisons in Figure 5. As shown in the images, our Hyden models scale well with higher input resolution and generate better geometry compared to baseline methods.

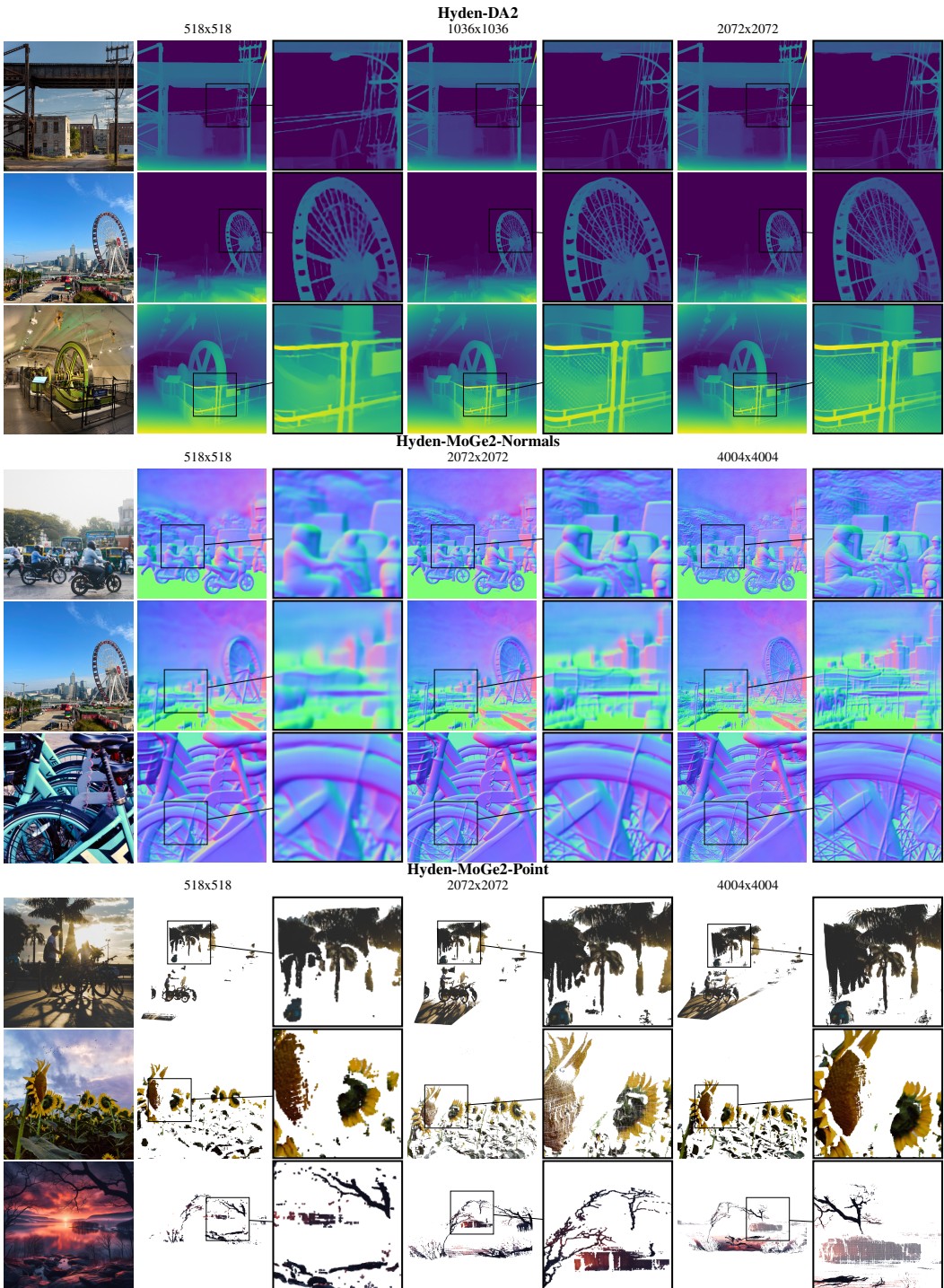

Figure 4: **Comparing Hyden models across different input image resolutions**

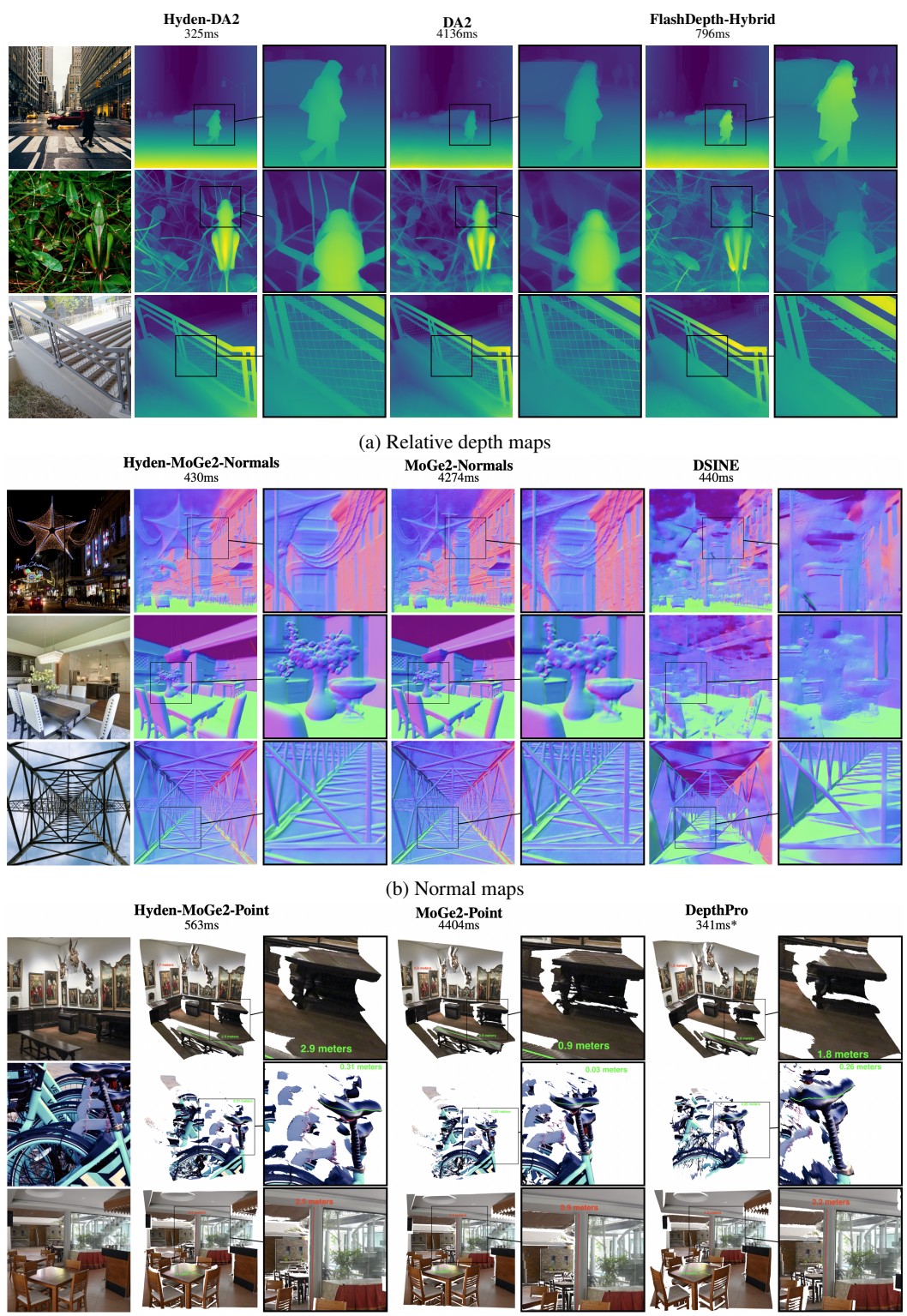

Figure 5: Additional qualitative comparisons