# OpenReview forum: "Hyden: A Hybrid Dual-Path Encoder for Monocular Geometry of High-resolution Images"
_ICLR.cc/2026/Conference — ICLR 2026 Poster_

### Official Review · Reviewer_R67Q · 2025-10-19

**Soundness:** 3
**Presentation:** 3
**Contribution:** 3
**Rating:** 6
**Confidence:** 5

**Summary:**

The paper presents a conv-transformer dual-path framework for high resolution depth estimation. The transformer processes lower resolution input whereas the conv processes higher one. After fusing the features, the model provides high resolution depth. Due to the scarcity of HR depth GT in the wild, the paper adopts a self-distillation method for labeling. Experiments show promising results.

**Strengths:**

- The dual-branch design is well-motivated and the overall framework is technically sound.

- The framework shows strong performance compared with current sota methods.

- The paper is well-writen and it's easy for me to follow the ideas.

**Weaknesses:**

- It would be better to add discussions about papers sharing the same ideas. For self-distillation: consider [1]. For dual branch, consider [2]. From my point view, discussing comparisons with [1] is kind of crucial as [1] was released 8 months ago and it was a trending paper. Though Hyden additionally presents a good architecture, the self-distillation is another major contribution claimed by the paper. [2] is an old paper but using the same dual-branch plus feature-level fusion idea. It would be good to add it in the related work.

    [1] Distill Any Depth: Distillation Creates a Stronger Monocular Depth Estimator
    [2] DepthFormer: Exploiting Long-Range Correlation and Local Information for Accurate Monocular Depth Estimation

- Since the scale and shift alignment is not perfect (is kind of noisy as far as I see), it would be better to present some clues about the error introduced by this alignment process, giving readers more insights about the upper bound of this method. It would benifit future potential follow-up works.

- One follow-up question: what if we use the way generating pseudo labels for direct evaluation? Given a HR image, we use DAV2/Moge for coarse prediction. Then, we crop patches from the HR image, use DAV2/Moge for patch-wise prediction. Finally, we align the patch-level prediction with coarse one and ensemble predictions to get final depth. I know this method leads to a super long inference time, but it would be good to have this one in the paper to indicate the performance and inference time gap.

- It's hard to reproduce the experimental results, majorly because the introduction to the training data is vague. Since the unlabeled data can be from any dataset, why don't use public available ones for the sake of potential follow-up works? Also, it would be good to include the results with different size of training data.

- I was wondering how many parameters are frozen vs, trainable in the architecture. Given a frozen transformer, I assume that only convolutional encoder and the fusion decoder are trainable.

**Questions:**

Please check the weakness.

---

> ### Author Response · Authors · 2025-11-18
> **Discussion about related works and paper reproduce**
>
> Thank you for the insightful review! Here are our responses:
>
> 1. Discussion about comparing with Distill Any Depth and DepthFormer: Thanks for pointing out the related works! For self-distillation, our local crop loss does share similar formulation with Distill Any Depth but our purpose is to solve the limited label issue for high resolution images while they are trying to improve the original monocular depth model at their native resolution. Therefore, we also explored the mixed-resolution training in the self distillation, which is proven to be very important based on Table 5 in ablation study. We also adopt a simple global alignment for our local crop labels and show its effectiveness across multiple downstream tasks: depth prediction, surface normal prediction and pointmap prediction, while Distill Any Depth studies different types of normalization for the relative depth task specifically. Additionally, Distill Any Depth actually proves its effectiveness for training with original ViT encoder and decoder while we prove the effectiveness for training with the additional CNN encoder and decoder only. The models we tested on are the two different branches of our proposed architecture.
>
>     For hybrid architectures, DepthFormer does propose a similar dual branch architecture combining CNN and ViT. However, our Hyden encoders are designed specifically for high resolution image inferencing at a relatively low time latency while DepthFormer was designed for how to better fuse CNN and ViT features. Although the design is promising, it still adds computation overhead. For example, their best model with swin-large encoder operates 1.5 times slower than DPT at 352×1216 resolution while our hyden encoder operates ~3 times faster than DPT at 2072 x 2072 resolution while maintaining better geometry prediction accuracy. Our proposed hyden model variants are much more efficient on high resolution image inputs.
>
> 2. scale and shift alignment: Thank you for this great question! During experimentation, we do find that after applying alignment, the geometry accuracy is significantly improved for most cases. To achieve the optimal results, we also apply a filter: if the local crop prediction disagrees with the global prediction on more than 50% of the total pixels in the target crop, this crop loss won’t be applied. For each task, we set up different thresholding methods on pixel prediction agreements. We will add more details in the supplemental material for this labeling step.
>
> 3. Patch-based refinement method comparison: Thank you again for this great question! We talk about the patch-based refinement methods in the related works and comparing to the SOTA method: PRO[1] in that branch, their method takes about 1.4 second for a native 2K resolution input while our Hyden-DA2 model takes about 0.1s. We will do a thorough quantitative comparison and add it in the supplemental material.
> [1]. One Look is Enough: Seamless Patchwise Refinement for Zero-Shot Monocular Depth Estimation on High-Resolution Images
>
> 4. Model reproduce: We will open source the model code and weights very soon. The reason we decided to use our own web crawled data is because of the limited high resolution images from the public dataset. While public datasets offer a wide variety of images, their resolution typically does not exceed 1k. And we are aiming for datasets with diverse high resolution images (around 2k resolution). We will run experiments and show the data scaling curve for different sized training data. Thanks for the suggestion!
>
> 5. Frozen vs trainable in the architecture: Yes, we freeze the vit encoders during training. And the CNN encoder only has 10M parameters and the decoders usually contain around 30~40M parameters. A vit-large encoder contain around 300M parameters. So in average, there are about 300M frozen parameters and ~40M trainable parameters.
>
> Please let us know if there are more questions!

---

> ### Comment · Reviewer_R67Q · 2025-11-26
> **Reply to Authors**
>
> Thanks the authors for these detailed replies. Most of my concerns are addressed.
>
> Though the authors point out that Distill Any Depth and the crop loss of Hyden have different targets, improving details for native resolution depth estimation and high-resolution images, those two targets are somehow similar. There is no design different from Distill Any Depth making it specific for this high-resolution target. This makes the tech contribution of this part weaker. The mixed-resolution training is hard to be claimed as a difference from my point view and it seems that this mixed-resolution training only appears in the ablation study section.
>
> I also see other reviewers have similar concerns about this issue. And this is one of two contributions claimed in this paper. Given this, I currently keep my score unchanged and see if concerns from nhZ4 and yCEW can be addressed.

---

> > ### Author Response · Authors · 2025-11-27
> > **Reply to Reviewer**
> >
> > Thanks for the feedback! We understand the rational behind your rating.
> >
> > We do want to point out again that the mixed-resolution self-distillation training is the key to improve the geometry accuracy at high-resolution image inferencing. As you can see from Table 1, the improvements are significant and no additional labels are needed. Although the technique is simple, it is highly effective.
> >
> > We will also improve our approach section so that it's clear that mixed-resolution training ensures the accuracy and local-crop training improves the prediction sharpness.

---

### Official Review · Reviewer_2xjn · 2025-10-28

**Soundness:** 3
**Presentation:** 4
**Contribution:** 3
**Rating:** 8
**Confidence:** 4

**Summary:**

This is a method for extending a dense predictor (based on ViT) to high resolution, applied to depth, surface normals and point maps prediction.  Given an already-trained network on a medium resolution (e.g. 518x518), the network is extended with a convnet (that scales linearly with image area) by fusing features from the original global network and the convnet.  This second-stage model is trained using a self-supervised method that predicts pseudolabels on crops using the original model, and aligns the crops to the global image prediction to form the training targets.

Notably, for any of the higher resolutions, the ViT's resolutions is held constant at 518.  This takes advantage of two properties/assumptions of the image captures: (1) the camera captures a scene framed by the image extent, so while increased resolution supplies more details, the world scene in the image doesn't change and can be handled by the ViT at fixed res; and (2) crops of the image are themselves scenes with smaller (or more distant) objects that are represented well enough in the original training set to be accurately predicted.

The method is incorporated into DepthAnything-v2 and MoGe2, with evaluations showing excellent improvements in latency vs resolution, accuracy measures and visual details at high res up.

**Strengths:**

This is an exceptionally simple method with great results, naturally combining the strengths of ViT and convnets to form a performant system.  The self-supervised training technique leverages existing models and data, making this extension mechanism fairly general.  I also found the presentation and writing very clear (except in a couple parts mention below), I much enjoyed reading it.

**Weaknesses:**

While this is an excellent extension method, it requires an already performant model to distill from.  And in particular, the original model must be performant at its full original (518) res on crops which may have a different scene framing or subjects from the original images.  While these are indeed the case for the applications here, it would be good to see this discussed as a requirement.

**Questions:**

To what extent do the ViT predictions of the global image vs crop disagree?  Some is of course inherently necessary, details and boundaries not present at lower res.  But are there times when the two scales disagree in larger areas?  How often does that occur and does it need to be handled in the alignment or by filtering?

Do you think this could be used for semantic segmentation and/or panoptic segmentation?  In particular, crops may or may not have enough of the objects present to identify them, depending on their size.  So the alignment between crops and global scale may need to account for that, as well as the possibility that the smaller crops' segmentations by the original model could output finer segmentation levels (e.g. car globally vs car/door/window in crops) depending on the model.

3.2.3:  I think this description is little confusing with the description of injecting into a support set, though I think I understand it.  Equivalently, though, is this the same as cropping the student SxS predictions and comparing against the corresponding teacher labels for the crop, summing over the crops in the task loss?

Fig 1 plots:  what are the stars in the plots?  also this plot is a little too small, I needed to zoom in a lot to read it.

---

> ### Author Response · Authors · 2025-11-18
> **Discussion about the local crop quality and semantic segmentation setup**
>
> Thank you for the insightful review! Glad you enjoyed reading the paper! Here are our responses:
>
> 1. Performant model pre-requisite: Yes, our extension framework does require a performant model to distill from. However, the CNN branch can take images from any resolution or aspect ratios. Since for feature fusion module, we use a bilinear interpolation to resize the feature map from the ViT encoder and that gives the model freedom to take in any scene framing images. Additionally for the global loss, we always make model predictions at the desired/targeted image resolution and then resize them to the original model resolution for loss computation. Loss at original resolution is to ensure the label quality but the model predictions can be designed or tuned for different resolutions/aspect ratios.
>
> 2. Local global disagreement: Thank you for this great question! During experimentation, we do find that after applying alignment, the geometry accuracy is significantly improved with local crop loss. To achieve the optimal results, we do apply a filter: if the local crop prediction disagrees with the global prediction on more than 50% of the total pixels in the target crop, this crop loss won’t be applied. For each task, we set up different thresholding methods on pixel prediction agreements. However, we do find that this filtering only marginally improves the model accuracy and it removes around 5 ~ 10% of the total crops. We will add more details in the supplemental material for this labeling filtering step.
>
> 3. Applying to semantic segmentation and/or panoptic segmentation: We thought about this task but didn’t try it. The main reason is that semantic segmentation is a high-level task and global scene-level features can be very important for some cases. For example, if the cropped images only contain parts of an object, it will be hard for the teacher model to infer its semantic class while for local geometry prediction, a depth or surface normal model doesn’t need the scene level feature to infer its low level geometry information. Only the scale of depth or global shift of surface normals need to be recovered.
>
> 4. Confusion on 3.2.3: Yes, it is exactly that “cropping the student SxS predictions and comparing against the corresponding teacher labels for the crop, summing over the crops in the task loss”. Sorry about the confusion. We will modify the text to make the description more clear.
>
> 5. Fig 1 plots: DepthPro can only be evaluated at 1536 resolution and it will be a point in the plots. That’s why we use stars to label each curve so that the DepthPro results can be visible. I do agree that the plots are small. We will arrange it a bit and make them significantly larger. Thanks for pointing this out!
>
> Please let us know if there are more questions!

---

### Official Review · Reviewer_yCEW · 2025-11-02

**Soundness:** 3
**Presentation:** 3
**Contribution:** 3
**Rating:** 6
**Confidence:** 4

**Summary:**

HYDEN is a high-resolution monocular depth, point map, and surface normal estimation model that achieves state-of-the-art accuracy with substantially reduced inference cost. Its architecture combines a low-resolution Vision Transformer (ViT) branch for global context with a full-resolution CNN branch for fine-grained details, and employs a self-distillation framework to address the scarcity of high-quality ground-truth supervision. Integrated into leading models such as DepthAnything-v2 and MoGe2, HYDEN delivers top performance on high-resolution benchmarks while providing significantly lower inference latency compared to competing methods.

**Strengths:**

- First of all, the key idea is interesting. The hybrid dual-path design effectively combines global ViT features and local CNN features without incurring the quadratic cost typical of pure ViT models. The modular encoder design makes HYDEN adaptable across multiple downstream tasks and backbones such as DA2 or MoGe2.
- Also, HYDEN achieves substantial speed-ups while preserving accuracy, making it practical for deployment in real-time or resource-constrained settings (e.g., robotic perception, self-driving planning etc.). This contrasts with prior methods that trade efficiency for high-resolution detail.
- Lastly, the model’s performance on zero-shot benchmarks highlights the robustness of the self-distillation framework and the hybrid encoder’s scalability.

**Weaknesses:**

- While the proposed framework is well designed and practically effective, its technical novelty is relatively limited. The core components such as the ViT–CNN fusion and pseudo-label-based self-distillation largely build upon existing approaches. The contribution primarily lies in the integration and scalability of existing techniques rather than introducing fundamentally new architectural or algorithmic innovations.
- Although the component-wise ablation studies are well organized, the paper lacks sufficient detail on the design rationale and training loss formulations for each module, making full reproduction of the method challenging.
- The effectiveness of the self-distillation pipeline relies heavily on the quality of the teacher model. However, the paper does not extensively discuss how weaker or biased teachers affect performance or convergence stability.

**Questions:**

- How sensitive is the self-distillation performance to the teacher model’s quality or prediction bias? Have the authors experimented with weaker teachers or ensemble teachers?
- Have the authors considered applying HYDEN to video sequences or multi-view inputs to assess temporal coherence in predictions?

---

> ### Author Response · Authors · 2025-11-18
> **Discussion about novelty, ablation study and self distillation framework**
>
> Thank you for the insightful review! Here are our responses:
>
> 1. Limited contribution: Our goal of this paper is to propose a flexible network design and an easy to implement training framework so that takes in any existing monocular geometry prediction model, we can easily modify the architecture and train it without additional supervision and it can adapt well to high-resolution image predictions with significantly lower inference speed. We proved this with three different downstream tasks and two different network architecture designs and we believe this extension design and training framework itself is an important contribution to this field.
>
>     The hybrid dual branch design was proposed in FlashDepth but we incorporate a smaller sized CNN instead of a vit-small model. We show faster inference speed at 4k resolution inferencing and most importantly, all our hyden variants outperform their base models significantly in terms of geometry accuracy while in FlashDepth, the geometry prediction accuracy of their hybrid model (FlashDepth-Full) was worse comparing to their base model (FlashDepth-L). One may argue that this gain might be from the high-resolution self distillation training, but even for evaluation at low resolution setting, the hyden variants do show gains at several benchmarks, such as NYU (Table 2 in supplemental), KITTI (Table 3 in supplemental), DIODE (Table 7 in supplemental), DDAD (Table 8 in supplemental). And we freeze our ViT encoders for the distillation training. Thus, we believe that CNN backbone itself offers complementary features to ViT encoders, which contributes to the consistent geometry accuracy improvements for Hyden models from different inference resolutions.
>
>      For self-distillation, our local crop loss does share similar formulation with Distill Any Depth but our purpose is to solve the limited label issue for high resolution images while they are trying to improve the original monocular depth model at their native resolution. Therefore, we also explored the mixed-resolution training in the self distillation, which is proven to be very important based on Table 5 in ablation study. We also adopt a simple global alignment for our local crop labels and show its effectiveness across multiple downstream tasks: depth prediction, surface normal prediction and pointmap prediction, while Distill Any Depth studies different types of normalization for the relative depth task specifically. Additionally, Distill Any Depth actually proves its effectiveness for training with original ViT encoder and decoder while we prove the effectiveness for training with the additional CNN encoder and decoder only. The models we tested on are the two different branches of our proposed architecture.
>
> 2. Detail on the design rationale and reproduction of the work: Thanks for pointing out the presentation issues in the paper. To improve the reproduction of this work, we plan to open source our model codes and weights soon (within one to two months). We will also add more training hyper-parameter details in the supplemental material for better reproduction.
>
> 3. Quality of the teacher model: Our goal of this paper is to propose a flexible network design and an easy to implement training framework so that takes in any existing monocular geometry prediction model, we can easily modify the architecture and train it without additional supervision and it can adapts well to high-resolution image predictions with significantly lower inference speed. So naturally, the teacher will just be the target model we want to add to our CNN branch. If the original model is weak or has bias, we do not address it in our paper since it is not our focus. We do acknowledge that this is one of the limitations of this paper. However, we did find out that with large-scale distillation, the student is able to predict smoother predictions compared to the original model, e.g. low light handling (Figure 4b second row: Hyden Moge2 vs Moge2) and blurring issues (Figure 5a second row in supplemental material: Hyden DA2 vs DA2).
>
> 4. Ensemble teachers: Thank you for this great question! We actually did try ensemble teachers with our training and we found that distilling from multiple teachers (even combining pseudo labels for one dataset and ground truth labels from other datasets), trained models turned to predict blurrier results compared to distilling from one source. We believe the key reason is that different teachers contain different model prediction bias, simply mixing the labels will cause the network to predict an averaged representation which is causing blurriness in the output.
>
> 5. Hyden to video sequences or multi-view inputs: Thank you for this suggestion! We haven’t tried it out but we are planning that for our future works.
>
> Please let us know if there are questions unanswered!

---

### Official Review · Reviewer_nhZ4 · 2025-11-03

**Soundness:** 3
**Presentation:** 3
**Contribution:** 2
**Rating:** 2
**Confidence:** 4

**Summary:**

This paper proposes a hybrid dual-path vision encoder for high-resolution geometry prediction. In addition, the authors employ off-the-shelf models to generate low-resolution depth for the entire image and high-resolution depth for local patches, which serve as the ground truth to supervise the network training. Experiments demonstrate that the proposed method achieves state-of-the-art performance and efficiency on the high-resolution geometry estimation benchmark.

**Strengths:**

1. The idea of hybrid dual-path architecture and self-distillation is simple and effective.

2.  The proposed method achieves state-of-the-art performance and efficiency.

**Weaknesses:**

1.  The novelty of the proposed hybrid dual-path encoder appears limited, as FlashDepth also adopts a similar methodology of fusing features from different resolution branches for high-resolution prediction. What is the ​fundamental difference​ in the encoder design between the proposed method and FlashDepth?

2.  The novelty of the proposed self-distillation also appears limited. Using cropping for self-distillation is also a common practice in self-supervised learning. In the field of depth estimation, Distill Any Depth [1] similarly employs a local-global distillation strategy based on cropping. A clear discussion of the core distinctions between the proposed method and Distill Any Depth is required.

3.  In the ablation study, the local crop loss is key to achieving fine-grained depth estimation. I'm curious to see what would happen if we only used the local crop loss and did away with the global loss altogether.

[1] Distill Any Depth: Distillation Creates a Stronger Monocular Depth Estimator, arXiv preprint arXiv: 2502.19204

**Questions:**

Please refer to Weaknesses.

---

> ### Author Response · Authors · 2025-11-18
> **Discussion about novelty and local crop loss**
>
> Thank you for the insightful review! For clarification, the key motivation of this paper is to improve the monocular vision models for high-resolution geometry prediction. We identify two main downsides of the current models: high inference latency and performance degradation with high resolution inputs.
>
> So we propose a flexible network design and an easy to implement training framework so that takes in any existing monocular geometry prediction model, we can easily add an extension network and train it without additional supervision so that the new model can achieve similar or better performance with high-resolution image inputs with significantly lower inference speed. We proved this with three different downstream tasks and two different network architecture designs and we believe this extension design and training framework itself is an important contribution to this field and then we discuss in details about the differentiations for each component:
>
> 1. Novelty of hybrid dual-path encoder: The hybrid dual branch design was proposed in FlashDepth but we incorporate a smaller sized CNN instead of a vit-small model. We show faster inference speed at 4k resolution inferencing and most importantly, all our hyden variants outperform their base models significantly in terms of geometry accuracy while in FlashDepth, the geometry prediction accuracy of their hybrid model (FlashDepth-Full) was worse comparing to their base model (FlashDepth-L). One may argue that this gain might be from the high-resolution self distillation training, but even for evaluation at low resolution setting, the hyden variants do show improvements at several benchmarks, such as NYU (Table 2 in supplemental), KITTI (Table 3 in supplemental), DIODE (Table 7 in supplemental), DDAD (Table 8 in supplemental). And we freeze our ViT encoders for the distillation training. Thus, we believe that CNN backbone itself offers complementary features to ViT encoders, which contributes to the consistent geometry accuracy improvements for Hyden models from different inference resolutions.
>
> 2. Novelty of self-distillation: For self-distillation, our local crop loss does share similar formulation with Distill Any Depth but our purpose is to solve the limited label issue for high resolution images while they are trying to improve the original monocular depth model at their native resolution. Therefore, we also explored the mixed-resolution training in the self distillation, which is proven to be the key to improve high resolution geometry prediction accuracy based on Table 5 in ablation study. We also adopt a simple global alignment for our local crop labels and show its effectiveness across multiple downstream tasks: depth prediction, surface normal prediction and pointmap prediction, while Distill Any Depth studies different types of normalization for the relative depth task only. Additionally, Distill Any Depth actually proves its effectiveness for training with original ViT encoder and decoder while we prove the effectiveness for training with the additional CNN encoder and decoder only. The two papers are actually training on two different branches of a hybrid dual branch encoder.
>
> 3. The local crop loss plays a crucial role in enhancing image sharpness for high-resolution predictions, as demonstrated in Table 4. However, cropping the image alters the camera intrinsics, which can introduce errors in global scale estimation. To address this, it is necessary to use global labels to accurately recover the true scale from local predictions. We experimented with training solely on refined local labels, but observed a significant decline in geometric accuracy. This suggests that local labels may introduce systematic bias due to changes in viewpoint, making it essential to retain a global loss component to ensure overall accuracy.
>
> Please let us know if there are questions unanswered!

---

### Meta-Review · Area_Chair_LNgG · 2026-01-07

**Summary:**

Most reviewers find the paper useful and technically sound after the rebuttal, but there is an ongoing disagreement about the level of technical novelty. Some reviewers argue that the design differences from prior work are not fundamental and therefore view the contribution as limited.

I believe the main contribution lies in clearly formulating the problem and proposing an effective and general framework to address it, which is validated across multiple tasks and models. Based on this, I recommend acceptance.

**Reviewer Concerns:**

The rebuttal addressed most of the technical and empirical concerns, including clarifying the architecture design, adding ablation studies, explaining the role of each component, and improving reproducibility and presentation. However, the concern about limited technical novelty — especially the similarity to prior work such as FlashDepth and Distill Any Depth — remains largely outstanding, as the rebuttal did not fully convince all reviewers that the contributions are fundamentally new rather than incremental.

**Reviewer Scores:**

Two reviewers initially gave acceptance scores (6) and did not raise major blocking concerns; their feedback was positive and would likely have remained unchanged after the discussion.

While the rebuttal addressed most technical and experimental concerns, it did not substantially change the reviewers’ perception regarding the limited novelty. As a result, the overall evaluation and score distribution would likely remain unchanged.

---

### Decision · Program_Chairs · 2026-01-26

Accept (Poster)